# Multi-century dynamics of the climate and carbon cycle under both high and net negative emissions scenarios

Charles D. Koven[1], Vivek K. Arora[2], Patricia Cadule[3], Rosie A. Fisher[4,5,6], Chris D. Jones[7], David M. Lawrence[4], Jared Lewis[8], Keith Lindsay[4], Sabine Mathesius[9,10], Malte Meinshausen[8,11], Michael Mills[4], Zebedee Nicholls[8,11,12], Benjamin M. Sanderson[4,5], Roland Séférian[13], Neil C. Swart[2], William R. Wieder[4,14], Kirsten Zickfeld[9]

[1]Climate and Ecosystem Sciences Division, Lawrence Berkeley National Laboratory, Berkeley, California, USA
[2]Canadian Centre for Climate Modelling and Analysis, Environment and Climate Change Canada, University of Victoria, Victoria, British Columbia, Canada
[3]IPSL, CNRS, Sorbonne Université, Paris, France
[4]Climate and Global Dynamics Laboratory, National Center for Atmospheric Research, Boulder, Colorado, USA
[5]Centre for International Climate and Environmental Research (CICERO), Oslo, Norway
[6]Évolution & Diversité Biologique, University of Toulouse Paul Sabatier III, Toulouse, France.
[7]Met Office Hadley Centre, Exeter, UK
[8]Climate and Energy College, School of Geography, Earth and Atmospheric Sciences, The University of Melbourne, Parkville, Victoria, Australia
[9]Department of Geography, Simon Fraser University, Burnaby, British Columbia, Canada
[10]now at Potsdam Institute for Climate Impact Research (PIK), Member of the Leibniz Association, Potsdam, Germany
[11]Climate Resource, Victoria, Australia
[12]International Institute for Applied Systems Analysis (IIASA), Laxenburg, Austria
[13]CNRM (Université de Toulouse, Météo-France, CNRS), Toulouse, France
[14]Institute of Arctic and Alpine Research, University of Colorado, Boulder, Colorado, USA

*Correspondence* to: Charles Koven (cdkoven@lbl.gov)

**Abstract.** Future climate projections from Earth system models (ESMs) typically focus on the timescale of this century. We use a set of five ESMs and one Earth system model of intermediate complexity (EMIC) to explore the dynamics of the Earth's climate and carbon cycles under contrasting emissions trajectories beyond this century, to the year 2300. The trajectories include a very high emissions, unmitigated fossil-fuel driven scenario, as well as a mitigation scenario that diverges from the first scenario after 2040 and features an 'overshoot', followed by a decrease of atmospheric $CO_2$ concentrations by means of large net-negative $CO_2$ emissions. In both scenarios, and for all models considered here, the terrestrial system switches from being a net sink to either a neutral state or a net source of carbon, though for different reasons and centered in different geographic regions, depending on both the model and the scenario. The ocean carbon system remains a sink, albeit weakened by carbon cycle feedbacks, in all models under the high emissions scenario, and switches from sink to source in the overshoot scenario. The global mean temperature anomaly is generally proportional to cumulative carbon emissions, with a deviation from proportionality in the overshoot scenario that is governed by the zero emissions commitment. Additionally, 23rd-century warming continues after the cessation of carbon emissions, in several models in the high emissions scenario, and in one model in the overshoot scenario. While ocean carbon cycle responses qualitatively agree both in globally integrated and zonal-mean dynamics in both scenarios, the land models qualitatively disagree in zonal-mean dynamics, in the relative roles of vegetation

and soil in driving C fluxes, in the response of the sink to $CO_2$, and in the timing of the sink-source transition, particularly in the high emissions scenario. The lack of agreement among land models on the mechanisms and geographic patterns of carbon cycle feedbacks, alongside the potential for lagged physical climate dynamics to cause warming long after $CO_2$ concentrations have stabilized, point to the possibility of surprises in the climate system beyond the 21st century time horizon, even under relatively mitigated global warming scenarios, which should be taken into consideration when setting global climate policy.

## 1. Introduction

Climate change is characterized by long timescales, associated with the accumulation of carbon in the atmosphere and other reservoirs of the Earth system due to emissions of $CO_2$ by anthropogenic activities, and the response of the climate system to the accumulated atmospheric $CO_2$ burden. The long lifetime of $CO_2$ in the atmosphere (Archer et al., 2009; Joos et al., 2013) and the proportionality between global warming and long-term cumulative $CO_2$ emissions are central features of the dynamics of the climate system (Matthews et al., 2009; Allen et al., 2009). These features underlie the widely used policy framework that proposes a 'budget' of remaining carbon emissions that would enable the climate system to remain below a given temperature (Rogelj et al., 2019). Future transient climate change scenarios using comprehensive Earth system models (ESMs) have typically focused on the timescale to the end of the 21st century, in order to inform near-term policy actions that may mitigate climate change. This end date of 2100 for these simulations has remained fixed, even though over 30 years have elapsed since the first IPCC assessment (IPCC, 1990).The start date of future scenarios has accordingly progressed from 1990 to 2015, shortening the length of these scenarios. Longer-term dynamics have been explored mainly using Earth system models of intermediate complexity (EMICs) (Zickfeld et al., 2013) and climate system emulators (Meinshausen et al., 2011, 2020; Nicholls et al., 2020b), which allow exploring such dynamics without the computational costs of resolving the full physical and biogeochemical dynamics of an ESM.  EMICs (and even more so emulators) typically represent land and ocean biogeochemical processes relevant to the long-term carbon cycle with less detail than comprehensive Earth system models, and so risk missing critical interactions and feedbacks. In contrast, ESMs have prioritized representing processes relevant on timescales to 2100, and may exclude or simplify processes important on longer timescales, such as permafrost carbon feedbacks on land, or sediment biogeochemistry in the ocean.

Initial studies using ESMs on this longer time horizon suggest that the proportionality of warming to carbon emissions that is both historically observed and projected on shorter timescales also holds on multi-century timescales, in unmitigated high-end warming scenarios (Randerson et al., 2015; Tokarska et al., 2016). It is less clear whether these relationships hold under mitigated or overshoot scenarios, where net negative carbon emissions are assumed later in the scenario, but the expectation is that the proportional relationship approximately holds for negative carbon emission as well (Zickfeld et al., 2016). Simple models show that the existence of a cumulative emissions to warming proportionality in such scenarios is sensitive to the

response timescales of physical and biogeochemical feedbacks in the Earth system (Sanderson, 2020). Existing experiments using ESMs and EMICs suggest that during a positive emissions phase, marine and terrestrial carbon cycles tend to absorb some fraction of added $CO_2$. During a removal phase, however, they tend to release $CO_2$ and thus partially offset the decline in atmospheric $CO_2$. As a result, we expect that under a scenario with positive emissions followed by net-negative emissions, warming remains approximately proportional to cumulative $CO_2$ emissions, but with an additional delay possible due to lags in the carbon cycle and thermal response to changing $CO_2$ (Tokarska and Zickfeld, 2015; Jones et al., 2016; Zickfeld et al., 2016; Tokarska et al., 2019).

To better understand the long-term dynamics of the carbon and climate systems, here we compare a set of five ESMs and one EMIC under a pair of high emissions and overshoot future climate scenarios that diverge in emissions in the mid-21st century, and explore the dynamics of carbon and climate under these contrasting trajectories. Further, because these models all report more detailed information that can allow some degree of process attribution to the dynamics, we separate the carbon cycle responses geographically, land from ocean, and on land we separate the soil and vegetation responses. We thus also explore whether and where the predicted carbon and climate responses are relatively robust, both within any one model over time and between scenarios, as well as across models for any given scenario and time period. This allows us to explore where model agreement does and does not exist, both in the globally integrated response, as well as in the regional and process drivers of that response. Where ESM or EMIC behavior shows either fundamental disagreement on geographic or process drivers for feedbacks, or shows global dynamics deviating significantly from the expected linearity between warming and cumulative emissions, we interpret it as showing a potential for surprises in the future dynamics of the Earth system.

## 2. Methods

### 2.1 Scenario Descriptions

All models were forced using the SSP5-8.5 and SSP5-3.4-overshoot scenarios (Riahi et al., 2017; Gidden et al., 2019; Meinshausen et al., 2020). These scenarios were constructed as part of the CMIP6 set of coordinated experiments for ESMs (Eyring et al., 2016), and arose out of the ScenarioMIP and SSP design effort (O'Neill et al., 2014, 2016) to cover a wide range of socioeconomic and policy scenarios and resulting trajectories of greenhouse gas forcings to the Earth system. The simple extensions beyond 2100 were adapted from those originally conceived in O'Neill (2016), as described in Meinshausen et al., (2020). Both of these scenarios follow the SSP5 21st century 'storyline' (Kriegler et al., 2017), which is premised on strong economic growth, relying largely on fossil fuels in the no-climate-policy baseline. However, they diverge in the year 2040: the SSP5-8.5 scenario continues on to long-term emissions growth, an 8.5 W m$^{-2}$ anthropogenic greenhouse gas radiative forcing by 2100, and $CO_2$ increases until the mid-23rd century. The SSP5-3.4 scenario dramatically changes course in 2040 after emissions peak, and continues with sustained net-negative $CO_2$ emissions until the mid-22nd century before stabilizing to near-zero emissions thereafter; the scale of negative emissions required for this overshoot (>5 PgC/yr for several decades) is much

larger than in the RCP2.6 scenario reported by Jones et al. (2016). These net negative emissions are largely driven by biomass energy with carbon capture and sequestration (BECCS), which in the land-use drivers of the scenarios is associated with a large conversion of pasture to crop lands (O'Neill et al., 2016). However none of the models explicitly track BECCS-related harvest fluxes.

All models were forced with specified atmospheric greenhouse gas concentrations (not greenhouse gas  emissions) (Meinshausen et al., 2020), and land use change forcings (Ma et al., 2020; Hurtt et al., 2020). In a concentration-forced ESM simulation, the land and ocean carbon cycles respond to the $CO_2$ concentrations in the atmosphere, which are specified via a global-mean timeseries (Fig. 1a), but do not feed back on atmospheric $CO_2$. We calculate compatible fossil fuel and industrial $CO_2$ emissions to satisfy the conservation of carbon within the Earth system by integrating the total atmospheric $CO_2$ reservoir, alongside the prognostic carbon reservoirs on land and ocean, such that anthropogenic fossil fuel and industrial emissions equal the sum of total carbon stock changes in land, atmosphere, and ocean (Liddicoat et al., 2021).  Land-use-driven carbon emissions are directly reflected in changes in the terrestrial carbon inventories and thus cannot be separately inferred based on terrestrial model dynamics themselves, as they are mixed with the model responses to changing climate and $CO_2$. For the analysis of global temperature change as a function of cumulative carbon emissions, we include in the emissions a land-use term in addition to the inferred fossil fuel term; this land use emissions term comes from the REMIND-MAgPIE IAM used to specify the SSP5-8.5 and SSP5-3.4-overshoot scenarios (Kriegler et al., 2017) as harmonized in Gidden et al., (2019), and thus does not differ between the models.  Because of differences in carbon cycle feedbacks and in the representation of land use fluxes, the $CO_2$ emissions inferred by ESMs to be consistent with a given $CO_2$ concentration pathway will not generally be equal to the $CO_2$ emissions that were provided by the IAM community for each scenario (Riahi et al., 2017; Gidden et al., 2019; Meinshausen et al., 2020), as shown below. The reason for this is that the representation of C cycle in ESMs is different from the model (MAGICC7) used to convert the IAM emissions into atmospheric $CO_2$ concentrations in the first place (Meinshausen et al., 2020). In the long-term extensions, land use is held constant after 2100, and the land-use fluxes used to calculate atmospheric $CO_2$ concentrations in the scenario specification go linearly from their 2100 value to zero at 2150, as described in Meinshausen et al., (2020).

Since the method for inferring compatible fossil fuel emissions from a concentration-driven ESM simulation is based only on conservation of mass, it is equally valid for net positive and net negative $CO_2$ emissions scenarios. However, if the ESMs disagree on the rate of land or ocean carbon uptake with the representation of land and ocean carbon uptake in MAGICC7 used to construct the $CO_2$ concentration timeseries, this disagreement will result in differences between the ESM-inferred and the scenario-specified $CO_2$ emissions. By comparing the ESM-inferred and scenario-specified emissions, we can determine whether any systematic differences between the ESM and MAGICC7 net carbon sinks exist.

## 2.2 Model Descriptions

Here we use results from five ESMs and one EMIC to explore the responses of the Earth system to the two long term scenarios. The models used here were the only models that had performed and archived the necessary experiments as of the time of writing. The five ESMs, all from the CMIP6 generation of models, are the Canadian Centre for Climate Modelling and Analysis fifth-generation Earth System Model, (CanESM5) (Swart et al., 2019d), Community Earth System Model, version two, Whole Atmosphere Configuration (CESM2-WACCM6) (Danabasoglu et al., 2020), the Centre National de Recherches Météorologique (CNRM) CNRM-ESM2-1 (Séférian et al., 2019), the Institut Pierre Simon Laplace (IPSL) IPSL-CM6A-LR (Boucher et al., 2020), and the U.K. Earth System Model (UKESM1) (Sellar et al., 2019). The EMIC is the University of Victoria Earth System Climate Model, version 2.10 (UVic-ESCM) (Mengis et al., 2020). Below we list some salient features of these models, and include more detailed model descriptions in Appendix A. In addition, further details on the ocean and marine biogeochemical components of these ESMs can be found in (Séférian et al., 2020; Arora et al., 2020; Canadell et al., 2021).

Of the models used here, there are several key differences in their land surface representations that may in principle govern the responses under these scenarios. Dynamic vegetation maybe particularly important both on longer timescales and in response to larger climate forcings, as ecosystems shift and reorganize in response to the changes; of the models here, only two (UKESM1 and UVic-ESCM) include a dynamic vegetation component while the rest assume fixed vegetation distributions. A terrestrial nitrogen cycle is particularly important in governing the response to both $CO_2$ and warming, as nutrients may limit the ability of plant productivity to increase under $CO_2$, and nutrient release due to warming soils may increase productivity; here the CESM2-WACCM6 and UKESM1 models both include a nitrogen cycle. A representation of carbon in permafrost layers may allow for large carbon releases from high latitudes in response to warming, and here two of the models (CESM2 and UVic-ESCM) include some representation of this process. Three of the models here (CESM2-WACCM, CNRM-ESM2-1, and UKESM1) distinguish between crop and pasture lands, which is relevant to the overshoot scenario and its large expansion of croplands from pasture.

We apply a seven-year running mean to all global timeseries in order to remove the short-term dynamics and focus on longer-term variability.

## 3. Results

### 3.1 Climate Responses

In the historical period and SSP5-8.5 scenario, global mean temperature change relative to the preindustrial (Fig. 1b) increases monotonically in all models, with a wide range of responses by 2300 from ~18 ºC in the CanESM5 to ~8 ºC in the UVic-ESCM model. Here, a notable difference arises between the ESMs and the UVic-ESCM EMIC, with much higher transient

warming in the ESMs than in the EMIC. This is at least in part due to a sampling bias related to the set of models that have performed these long-term scenarios: four of the five ESMs used here report transient climate response (TCR) greater than 2.3 °C and a transient climate response to emissions (TCRE) greater than 2 °C/EgC (mean of 2.16 °C/EgC) versus the CMIP6 mean of 2.0 °C TCR and 1.8 °C/EgC TCRE (Arora et al., 2020); whereas the specific version of UVic-ESCM used here reports a TCR of 1.8 °C and a TCRE of 1.8 °C/EgC (MacDougall et al., 2020), closer to the CMIP6 mean. The one ESM with lower sensitivity, CNRM-ESM2-1, reports a TCR of 1.84 °C and a TCRE of 1.63 °C/EgC (Arora et al., 2020). During the period of $CO_2$ stabilization and decline in the 23rd century, four of the ESMs continue to warm substantially, while in one ESM (UKESM1) and the EMIC, the global temperature stabilizes. Since these are concentration-forced experiments, this divergence in long-term warming after stabilization of $CO_2$ concentration implies a substantial slow component to the physical climate feedback in the models that continue to warm, beyond the effective transient values reported above, which reflect short-to-medium term feedback processes that dominate the TCR (and implicitly the TCRE) (Proistosescu and Huybers, 2017).

In contrast, in the SSP5-3.4-overshoot scenario, global temperatures follow the $CO_2$ concentration trajectory to first peak and then cool during the 21st century in all models. Subsequent dynamics vary between the models: most stabilize at a cooler temperature than the peak 21st century value, while one model (CESM2-WACCM) reaches a minimum temperature at ~2200, and then resumes warming, albeit at a slower rate, during the 23rd century. As in the very high emissions scenario, there is a separation in the amount of warming between the relatively less sensitive EMIC and more sensitive ESMs both at the peak and in the subsequent overshoot and stabilization period.

### 3.2 Carbon Cycle Responses

Responses of the globally-integrated terrestrial and marine carbon cycle to the two scenarios for all models are shown in fig. 1c-d, and reported in table B1. Under both the SSP5-8.5 and SSP5-3.4-overshoot scenarios, the terrestrial carbon cycle (Fig 1c) in all models shifts at some point from being a net sink of carbon from the atmosphere to a neutral or net source of carbon to the atmosphere. In the SSP5-8.5 scenario, the timing of this transition varies widely between models, from ~2100 in UVic-ESCM to ~2220 in CESM2-WACCM. The magnitude of the carbon fluxes also varies widely between models, with CanESM5 showing strongest terrestrial uptake, peaking around 2100, and then reversing to become the strongest terrestrial carbon source out of the models examined here during the 23rd century. Model spread of the land sink increases substantially from the 21st to the 22nd century in the SSP5-8.5 scenario, as indicated by the increasing standard deviation across the ensemble of cumulative sink from 264 ±172 Pg C for the period 2015-2100 to -29 ±264 Pg C for the period 2100-2200, shown in table B1.

Overall, the pattern of terrestrial sink-to-source transition under long-term high emissions is qualitatively consistent with the results of Tokarska et al. (2016), which show a similar transition in all of the models examined in the RCP8.5 extension experiment. This pattern follows from the dynamics described by Randerson et al. (2015) whereby terrestrial carbon-climate feedbacks strengthen over time, at the same time that the terrestrial carbon-concentration feedbacks weaken, although the

experimental protocol followed here, which does not separate $CO_2$ climate and physical effects as in (Arora et al., 2020), does not allow this feedback decomposition to be performed.

For the SSP5-3.4-overshoot scenario, model agreement of the terrestrial carbon cycle is much higher, with all models transitioning from sink to source during the late 21st or early 22nd centuries, which counteracts some of the net-negative anthropogenic emissions by that time in terms of their effect on lowering atmospheric $CO_2$ concentrations. The ensemble spread in cumulative carbon uptake also narrows from the 21st century (146 ±78 Pg C) to the 22nd century (-60 ±48 Pg C). This change in sign is consistent with the CMIP5 RCP2.6 results shown in Jones et al. (2016). The timing of the biospheric

switch from sink to source follows by decades the change in the sign of $CO_2$ emissions from net positive to net negative. All of the models then revert to a roughly carbon-neutral terrestrial biosphere during the 23rd century. Notably, models across the ensemble show a reduced range of variation in the magnitude of carbon fluxes for the SSP5-3.4-overshoot scenario, relative to the SSP5-8.5 scenario.

Over ocean (Fig 1d, table B1), inter-model agreement is in general much higher than over land, although ensemble spread does increase beyond 2100 in the SSP5-8.5 scenario, from a cumulative uptake of 392 ±31 Pg C in the period 2015-2100 to 445 ±71 Pg C in the period 2100-2200. Peak carbon uptake for both scenarios occurs prior to 2100 in all models, with an earlier and smaller-magnitude peak in the SSP5-3.4-overshoot than the SSP5-8.5 scenario. In the models, the ocean carbon uptake then gradually weakens but remains positive through the 22nd and 23rd centuries in the SSP5-8.5 scenario, while in the SSP5-

3.4-overshoot scenario, uptake rapidly reverses to become a source through most of the 22nd century (lagging behind the change in the sign of net $CO_2$ emissions by decades), before then reversing again in the late 22nd century to become a weak sink again through the remainder of the scenario.

**3.3 Diagnosed $CO_2$ Emissions**

Annual (fig. 2a) and cumulative (fig. 2b, table B1) fossil fuel $CO_2$ emissions, which are compatible with the specified $CO_2$

concentration pathway in these simulations, follow the overall trajectory of the fossil fuel emissions used to generate the concentrations scenario using the MAGICC7 model, as was also found by Liddicoat et al. (2021) for the full set of SSP scenarios through 2100. The additional spread in ESMs and the UVic EMIC is due to the difference between the models' carbon cycles and the carbon cycle in the MAGICC7 model (section 2.1).

In the SSP5-8.5 scenario, the model ensemble spread in compatible emissions is widest at the end of the 21st century when emissions also peak, and declines during the 22nd century. In one model under SSP5-8.5 (CanESM5), negative emissions are required in the 23rd century to balance the strong and sustained terrestrial carbon source active at that time in that scenario, whereas in the rest of the models slightly positive or roughly zero emissions are inferred for the scenario. In the SSP5-3.4-overshoot scenario, the ensemble spread in compatible emissions peaks first at the time of peak positive $CO_2$ emissions, and

then increases again during the period of strongest negative $CO_2$ emissions, as models disagree on the magnitude of carbon cycle responses to each of these phases.

The shape of the cumulative diagnosed $CO_2$ emissions (fig. 2b) roughly follows the trajectory of the atmospheric $CO_2$ concentrations shown in fig. 1a. Ensemble spread in cumulative diagnosed $CO_2$ emissions shows the relative responses of the

carbon cycles in each model to positive and negative $CO_2$ emissions, with, e.g. the IPSL-CM6A-LR model requiring higher cumulative emissions to balance its stronger sink throughout the entirety of the SSP5-3.4-overshoot scenario, and the CanESM5 model requiring higher amounts of cumulative $CO_2$ emissions to balance its high sink in the SSP5-8.5 scenario until ~2200, when that model's terrestrial system reverses from a strong sink to a strong source.

Each of the fluxes, averaged across the models, are shown together in figure 2c-d. Here, for each scenario, the pink line shows the inferred emissions time series, the black line shows the change in atmospheric $CO_2$, and the accumulation in land (green) and ocean (blue) are shown by the area of sinks (hatching) or sources (stippling). This shows lags in the land and ocean carbon fluxes in response to changes in emissions, particularly for the SSP5-3.4-overshoot scenario (fig. 2d), where the terrestrial and ocean systems remain sinks for several decades during the period of declining and negative $CO_2$ emissions, before they switch

to become sources, which partially offset the negative emissions. In the SSP5-8.5 scenario, lags are less evident, but the net behavior of the ocean is to at least partially offset the net carbon losses on land during the period after the mid-22nd century.

### 3.4 Temperature Response to Cumulative Emissions

Plotting global mean temperature change as a function of diagnosed cumulative $CO_2$ emissions (fig. 3) reproduces the near linear relationship between temperature change and cumulative $CO_2$ emissions described in (Tokarska and Zickfeld, 2015;

Jones et al., 2016; Zickfeld et al., 2016). Note, however, that the temperature change shown here includes the response to non-$CO_2$ forcings, whereas the linear relationship is strictly defined only for $CO_2$ (Matthews et al., 2009; Collins et al., 2013), and thus the relationship shown here represents an "effective TCRE" (Matthews et al., 2017) that includes these non-$CO_2$ forcings. Further, following, e.g., Canadell et al. (2021), we add an estimated land use $CO_2$ flux from the IAM-derived scenario specifications to the diagnosed fossil $CO_2$ emissions for each model.


There is some deviation from linearity in the cumulative carbon emissions to temperature relationship in SSP5-8.5 scenario. Initially, up to approximately 2000 PgC, temperature increases less than linearly with cumulative $CO_2$ emissions. There are two potential explanations for this curvature. The first potential explanation is the role of non-$CO_2$ forcers, which contribute a larger fraction of the total greenhouse gas forcing in the early than late part of this scenario (fig. C1). The second potential

explanation is lags in the carbon and climate systems relative to emissions. An analysis with $CO_2$-only experiments up to 2100 (Nicholls et al., 2020a) found a similar slight negative curvature as observed here, suggesting that lags in the carbon and climate systems are dominant up to around 2000 PgC. The temperature vs cumulative emissions relationship is approximately

linear for cumulative emissions between 2000 PgC and 4000 PgC, except for the UVic ESCM. The less-than-linear relationship in EMICs was noted before and attributed to more efficient ocean heat uptake and/or a stronger saturation of $CO_2$ radiative forcing at high cumulative emissions (Herrington and Zickfeld, 2014; Tokarska et al., 2016). In the final half-century of the SSP5-8.5 scenario, temperature in the ESMs continues to increase in response to approximately stable radiative forcing. This continued warming reflects the lags in the carbon and thermal response to $CO_2$ emissions and non-$CO_2$ forcings. The one EMIC shows a more linear response in this part of the scenario than the ESMs. In the ESMs shown here for the SSP5-8.5 scenario, this lagged-warming tail is larger, particularly in the case of CanESM5, than the corresponding behavior shown in Tokarska et al., (2016).

By breaking the cumulative emissions plots into roughly centennial-length segments, figure 3 shows the dynamics for the two scenarios over time for all of the models. This underscores the continuity of the cumulative emissions curve through the 22nd century in the SSP5-8.5 scenario (fig. 3a), and the break in that relationship for several of the models during the 23rd century. For the SSP5-3.4-overshoot scenario (fig. 3b), the separation by centuries further shows the slight nonlinearity evident in some—but not all—of the models during the peak and initial overshoot period. In the overshoot scenario, there is not a consistent deviation from linearity at the point of overshoot and negative $CO_2$ emissions. Some models (UVic-ESCM, CanESM5) follow roughly the same trajectory in temperature vs cumulative emissions space in the initial period of negative emissions, while others (IPSL-CM6A-LR, CESM2) follow a lower-temperature trajectory after peak warming, and one model (UKESM1) follows a higher-temperature curve in temperature vs cumulative space after peak emissions. CESM2-WACCM also shows a distinctly different 23rd-century response to the other models, with a significant increase in temperatures in response to near-constant radiative forcing, and nearly zero inferred emissions, over this period.

To further understand why some models show a higher and some a lower temperature for a given amount of cumulative emissions in the negative emissions period, we first identify a metric of this overshoot asymmetry. Here we use the point of 200 Pg C below the peak cumulative emissions for each model, and calculate the asymmetry as the difference of the descending (negative-emissions period) minus the ascending (positive-emissions period) temperatures for 20-year periods centered at this point. We hypothesize that the asymmetry could be due to different roles of carbon versus thermal inertia in the declining $CO_2$ phase (Boucher et al., 2012; Zickfeld et al., 2016), and thus related to the zero emissions commitment (ZEC), which is the temperature change that occurs after reaching zero $CO_2$ emissions. Values of ZEC from the ZECMIP CMIP6 experiment are reported by MacDougall et al. (2020) for all models here except for IPSL-CM6A-LR; also CESM2 in MacDougall et al. (2020) was not run with the full upper atmosphere (WACCM) configuration as it was here, however we do not expect that difference to strongly affect this comparison. The comparison of the overshoot asymmetry metric and the 90-year ZEC values for each model is shown in figure 4; the correlation is high ($r^2$=0.96) and the best fit regression line is near 1:1. Comparison against the 50-year ZEC from MacDougall et al. (2020) is similar, with the $r^2$ only slightly reduced to 0.93. This supports the idea that the overshoot asymmetry here and the ZEC are governed by the same processes.

As these scenarios are concentration-driven rather than emissions-driven, the uncertainty due to carbon cycle processes shows up in figure 3 as a spread in cumulative $CO_2$ emissions (horizontal axis) between ensemble members, rather than a vertical divergence as it would appear in an emissions-driven scenario. However, the self-consistency between the climate and carbon cycles that results from the inferred-emissions approach, as well as the qualitative consistency between the models and the emulator that was used to translate scenario fluxes to atmospheric $CO_2$ concentrations in the scenario specification, together ensure that the behavior will be similar between concentration-driven and emissions-driven dynamics, even under these extreme scenarios with either very high or net negative emissions. The consistency between the model dynamics that are concentration-forced here and those of the emissions-forced runs from ZECMIP (MacDougall et al., 2020) further supports the argument that temperature-cumulative emissions relationships between concentration-forced and emissions-forced experiments are comparable even under strong net negative $CO_2$ emissions.

### 3.5 Regional variation in carbon and temperature dynamics

### 3.5.1 Terrestrial carbon cycle

Aggregated globally, there is some commonality and a large degree of divergence between models across these two contrasting scenarios. While all models show some consistent patterns (e.g., a shift on land from sink to source), individual models also show differing dynamics both in the patterns, the timing, and the magnitudes of the carbon and temperature response. It is possible to disaggregate these dynamics regionally to better understand the mechanistic basis of the carbon and temperature response, and to explore whether any qualitative similarity holds at the more regional scales. We thus focus on zonal-mean trajectories of carbon and temperature as a way to further understand the degree of similarity in results across models and within any model over time and scenarios.

Figure 5 shows zonal-mean terrestrial carbon flux dynamics for the five models and two scenarios over the full historical to future period. The value for a given latitude is the average over all land cells in that latitude, regardless of the fractional coverage of land in grid cells. In the historical and near-future (prior to 2040) time period that are shared between the scenarios, the five models already show a strong divergence in behavior: CanESM5 projects 21st century carbon sinks in both the tropics and northern high latitudes; CESM2-WACCM has one main sink area in the tropics and a much weaker sink at northern mid and high latitudes; IPSL-CM6A-LR and UKESM1 also have one main sink area, but in the northern mid- and high latitudes, and UVic-ESCM shows a weak sink in the tropics and growing carbon source in the higher latitudes. Over time in the SSP5-8.5 scenario (fig. 5a), each of these models show further divergent results: in CanESM5, both the tropical and northern high latitude regions shift from sinks to sources at roughly the same time, becoming sources by the mid-22nd century; in CESM2-WACCM, the tropics remain a sink through the end of the 22nd century, while the northern high latitudes shift to become a source by the end of the 21st century, with the source peaking during the 22nd century and weakening thereafter; in IPSL-

CM6A-LR the northern sink weakens gradually over time to become neutral by the mid-22nd century, while the tropics become

a strong source of carbon during the 22nd century; in UK-ESM1, the northern sink is sustained while the tropics shift to become a source, and in UVic-ESCM the northern high latitudes become a strong source and the tropics a weak source. Thus the regionally-disaggregated dynamics show even greater divergence than the global integral, with differing locations—and thus mechanisms—driving the overall shift from sink to source across models. Further, the areas of most active terrestrial carbon cycle dynamics shift from one region to the next across centuries within any one model.


Zonal-mean dynamics are both more muted in magnitude and more similar between models for the SSP5-3.4-overshoot scenario (fig. 5b). In both the CanESM5 and CESM2-WACCM models, the early sinks weaken in favor of a source of carbon in the tropics during the net negative $CO_2$ emissions period from roughly 2050 to 2150. The IPSL-CM6A-LR and UKESM1 models show similar dynamics, but with a larger overlay of interannual variability. The UVic-ESCM model shows a relatively

brief but strong loss of carbon from northern high latitudes during the period of peak warming, and a slower and weaker loss of carbon from the tropics during the subsequent period of net negative $CO_2$ emissions.

Further disaggregating the dynamics into zonal-mean vegetation and soil carbon pools (fig. 6) shows even greater divergence between the models. Vegetation carbon pools accumulate in both the tropics and northern mid-high latitudes in the SSP5-8.5

scenario in both CanESM5 and CESM2-WACCM; in IPSL-CM6A-LR, the northern latitudes gain vegetation carbon but the tropical latitudes lose large amounts of vegetation carbon; in UKESM1 vegetation carbon accumulates at mid-high latitudes of both hemispheres but is roughly neutral in the tropics; in UVic-ESCM, northern vegetation is a weak sink and tropical vegetation is roughly neutral. For soils, CanESM5 gains carbon in both the tropical and mid-high latitude belts, albeit with a delay relative to vegetation pools, through the mid 22nd century, but then shifts to lose carbon in soils from both belts by the

end of the 23rd century; CESM2-WACCM gains soil carbon through most of the world but also projects substantial carbon losses from the northern high latitude soils beginning in the late 21st century; IPSL-CM6A-LR loses soil carbon, mainly from the tropics, starting mainly during the 22nd century; UKESM1 shows a stronger tropical soil carbon loss and a higher latitude soil carbon gain; and UVic-ESCM shows strong losses of carbon at northern high latitudes and gains of soil carbon in the northern mid-latitudes. Thus, under the SSP5-8.5 scenario, both soil and vegetation dynamics differ markedly across the

models, as well as regionally within each model.

For the SSP5-3.4-overshoot scenario, zonal-mean disaggregation of vegetation and soil carbon (fig. 6c-d) shows some greater degree of similarity between model dynamics. All five models agree that northern mid-high latitudes would gain carbon in vegetation in this scenario. In the tropics, three models (CanESM5, CESM2-WACCM, and IPSL-CM6A-LR) predict that

carbon gains in tropical vegetation peak by the end of the 21st century, while UKESM1 projects sustained tropical vegetation carbon losses from the historical through the end of the scenario, and UVic-ESCM shows more neutral behavior of vegetation globally. CESM2-WACCM and UKESM1 also show substantial losses of vegetation carbon in subtropical ecosystems. For

soil carbon dynamics, the patterns are much more muted than in the high-emissions case, with weaker but sustained carbon gains in soils of northern high latitudes in the CanESM5, IPSL-CM6A-LR, and UKESM1 models, and a weaker loss of carbon

from northern high latitudes and gain of carbon in the northern mid-latitudes in the CESM2-WACCM model, and for the UVic-ESCM model, northern soil carbon losses are weaker than in the very high emissions case but still stronger than any of the other models.

### 3.5.2 Ocean carbon cycle

Zonal-mean breakdowns of the ocean carbon cycle are much more consistent between models (fig. 7). All models show near-

term sinks in the mid and high latitudes of both hemispheres, with sources in the tropics. Under the SSP5-8.5 scenario, all models show a poleward migration of the Southern Ocean sink, and a weakening followed by a strengthening of the tropical source. In five of the six models, the northern mid- and high- latitude sinks weaken during the 22nd century, while they remain strong in the IPSL-CM6A-LR model. While the zonal-mean patterns of the ocean carbon flux are broadly consistent across the models, the magnitudes and meridional extents of the source and sink regions vary significantly, leading to the large spread

in net global fluxes across models seen under the SSP5-8.5 scenario (Fig 1d). In the SSP5-3.4-overshoot scenario, all models show roughly similar dynamics: the tropical source strengthens, the northern mid- and high- latitudes sinks weaken, and the southern ocean shifts from sink to source, although differences in the timing, strength and meridional extent of these transitions are again evident between models.

### 3.5.3 Distribution of ensemble-mean land and ocean carbon changes

Spatial patterns of the ensemble-mean time-integrated carbon changes over both land and ocean (fig. 8) exhibit some consistent patterns across the ensemble over successive periods of time; hatching in the figure is indicated where two or more of the models disagree in sign with the ensemble mean. During the historical period, models agree on a carbon sink in the tropical forests of all three continental regions, as well as a sink in the northern mid-high latitudes, and an ocean sink in most regions with a higher sink strength in the North Atlantic and Southern Ocean.


Under the SSP5-8.5 scenario, for the 21st century, tropical forest sink strength is projected to increase in the ensemble mean, but the area of model agreement decreases relative to the historical period. Boreal forest sink strength also is projected to increase strongly in the 21st century under SSP5-8.5, with high model agreement on sign. In the 22nd century under SSP5-8.5, South American and African tropical forest regions switch from sink to source in the ensemble mean, with high agreement,

while southeast Asian tropical forests remain a sink in the ensemble mean, but with low model agreement; high latitude terrestrial regions also lose any consistent signal. In the 23rd century under SSP5-8.5, South American and African tropical forests continue as sources, with the African tropical forest region becoming a stronger source than the South American region, and the Asian forest region now also switches from sink to source in the ensemble mean. Overall ocean sinks strengthen,

particularly in the Southern Ocean, from the 21st century to the 22nd, and stay roughly constant into the 23rd, while the North
Atlantic sink weakens and becomes a slight source by the 23rd century.

Under the SSP5-3.4-overshoot scenario, 21st century integrated uptake is weaker in the Amazon forest region, southeast Asian tropical forests, and northern mid-high latitude forests, while the African tropical forest region acts as a source. In the 22nd century under SSP5-3.4-overshoot, all tropical forest regions transition from sink to source, and model agreement elsewhere
is low. In the 23rd century under SSP5-3.4-overshoot, the entire land surface has a roughly neutral carbon balance, with low agreement on sign. The ocean carbon cycle acts as a progressively weakening sink from one century to the next in the SSP5-3.4-overshoot scenario.

### 3.5.4 Temperature

Regional temperature dynamics are roughly similar between ESMs (fig. 9). All models show polar amplification, and thus
warming proceeds faster at high latitudes, particularly in the northern hemisphere. Under the high emissions scenario, global warming is overwhelming, with >10 degrees C warming at all latitudes and much higher warming at the poles, with warming reaching 10 degrees at the northern polar region within this century in all five models. Under the overshoot scenario, polar amplification is still present in all models, and global warming peaks and then declines and stabilizes after the peak $CO_2$ period in all models except CESM2-WACCM.


In CESM2-WACCM, for the overshoot scenario, the northern mid- to high latitudes return to almost the preindustrial temperature during the 22nd century, and then subsequently warm again in the 23rd century, despite no further $CO_2$ concentration increases; this area is responsible for the vertical tail to the cumulative emissions-temperature change plot in the 23rd century in that model (Fig. 3b). Plotting the 100-year mean temperature difference between the 23rd and 22nd centuries
(fig. 10a), shows that the 23rd century warming in the model is centered on the Northern Atlantic, suggesting a control by Atlantic Meridional Overturning Circulation (AMOC). To explore this hypothesis, we calculate AMOC as the maximum value of the annual-mean meridional volume flow stream function in the Atlantic basin north of 20N, and plot time series of this for all ESMs and scenarios in figure 10b. This shows that CESM2 AMOC starts out with stronger AMOC than the other ESMs, which substantially weakens during the period of warming to reach a minimum at around 2100 and recovers thereafter in the
SSP5-3.4-overshoot scenario, with a much stronger rebound than other ESMs considered here. This AMOC recovery response is consistent with earlier long-term overshoot scenarios (Nakashiki et al., 2006) as well as long-term constant 2xCO$_2$ experiments (Manabe and Stouffer, 1994). Thus this supports the interpretation that the AMOC recovery in that model drives the 23rd-century warming. An additional piece of evidence to support the transient weakening and subsequent recovery of AMOC as being the key driver of the CESM-WACCM temperature vs cumulative emissions nonlinearity shown here comes
from the comparison of Hu et al. (2020) between CESM2 and a closely related model, E3SM. They show that both models have similar ECS but CESM has a substantially lower TCR, which they attribute to its higher sensitivity of AMOC strength to

warming. The 23rd century warming in CESM2-WACCM thus appears to reflect an AMOC that is transiently weakened during the 22nd century due to freshwater influx associated with warming, leading to relative cooling around the North Atlantic and throughout the high latitudes, but which then recovers and removes that cooling anomaly that was present during the weakened-AMOC period.

## 4. Discussion

The concept of proportionality of global warming to cumulative emissions and the related metric of transient climate response to cumulative $CO_2$ emissions (TCRE) are enormously valuable in understanding the expected response of global temperature change to anthropogenic emissions. At the same time, the utility of this framework is limited by both the persistent spread in TCRE across model ensembles (Arora et al., 2020; Jones et al., 2013), as well as the possibilities of behavior in the coupled climate and carbon cycle systems that may give rise to nonlinear trajectories of temperature as a function of cumulative $CO_2$ emissions. Recent IPCC reports (Canadell et al. 2021; Rogelj et al., 2018) use a framework described in (Rogelj et al., 2019) to identify the remaining carbon budget consistent with stabilization of global temperatures at or below a given level. This abstraction of the climate system allows for two additional terms beyond the TCRE: a zero emissions commitment (ZEC), which is any warming which arises after the point that $CO_2$ emissions reach net zero, and thus would lead to vertical tails (either positive or negative) in the cumulative emissions plots (although part of the tail warming shown here may be in response to near-constant non-$CO_2$ forcing), and an allowance for Earth system feedbacks that are unrepresented or underrepresented in existing Earth system models and thus not included in the spread of TCRE from ESMs. Physical or biogeochemical lags in the Earth system, beyond those quantified by the ZEC or specifically enumerated as unrepresented feedbacks, are not accounted for in the Rogelj et al., (2019) framework, though the updated framework of Nicholls et al., (2020a) allows for non-linearities between cumulative $CO_2$ emissions and $CO_2$-induced warming in remaining carbon budget calculations. Longer-duration and overshoot scenarios may be useful in identifying whether further complexity in the relationship between global temperature and cumulative $CO_2$ emissions exists and should be considered in remaining carbon budget or other policy frameworks.

Here we show that, for overshoot scenarios such as SSP5-3.4-overshoot, the ZEC also governs the degree of temperature asymmetry at a given cumulative emissions level between the negative and positive emissions periods (fig. 4). This result can be understood as the ZEC being a general measure of the relative strength of lagged warming versus lagged $CO_2$ uptake at longer timescales, which occur both if emissions are zero or if they become negative. Thus the ZEC represents an additional committed temperature change that must be factored into carbon budgets, whether or not an overshoot in $CO_2$ emissions occurs. The IPCC AR6 assessed the magnitude of the ZEC as being approximately zero with a 1-sigma range of ±0.19 (Canadell et al., 2021; Lee et al., 2021), thus representing an important uncertainty in the remaining carbon budget. The result here, that the ZEC governs a wide range of dynamics from zero emissions to net negative emissions on these longer timescales emphasizes the importance of better constraining the magnitude of the ZEC and understanding its distinct mechanistic drivers.

The pair of scenarios explored here bracket a wide range of possible dynamics in the Earth system over the next few centuries, from a high $CO_2$ concentration world with continuous and overwhelming global warming over the coming centuries to one in which $CO_2$ is stabilized and reduced following a peak warming during this century (Figs. 1-2). In each of these scenarios, the models studied in general follow the expected linearity in TCRE, before 2200 (Fig. 3). At the same time their internal dynamics vary widely from each other, particularly in the terrestrial carbon cycle and under high levels of global warming, where there

is little agreement on the geographic and mechanistic drivers of the terrestrial carbon cycle responses to the warming (Figs. 4-5). Possibly this is due to some degree of tuning (either implicit or explicit) to capture the observed globally-integrated 20th century carbon balance trajectory, a constraint whose influence weakens at regional levels and over time into the future (Hoffman et al., 2014).

The five models used here vary widely in the representation of their terrestrial biospheres: two (UKESM1, and UVic-ESCM) include vegetation dynamics, while the other three (CanESM5, CESM2-WACCM, IPSL-CM6A-LR) use prescribed and static distributions of plant functional types. Two models (CESM2-WACCM and UVic-ESCM) include the dynamics of deep and frozen soil carbon, while the others drive soil biogeochemistry using near-surface soil temperatures and thus exclude the possibility of permafrost carbon feedbacks to climate change. Two models (CESM2-WACCM and UKESM1) include the

nitrogen cycle on land, while the others do not. While there does not appear to be a general signature associated with the inclusion of vegetation dynamics or nitrogen here, the inclusion of permafrost carbon in both models here does lead to a signature of large soil carbon losses at high latitudes under the high-warming scenario. Overall, the models differ widely in the aggregated magnitude of their responses to climate and elevated $CO_2$ (Arora et al., 2020). The structural differences likely underlie the diversity of global and regional responses, although given the myriad structural and parametric differences

between the models it is not possible to attribute the dynamics in a more rigorous way (Fisher and Koven, 2020). It is also not possible with the limited sample size considered here to assess whether model agreement in shorter term response arises due to common representation of relevant processes or calibration constraints imposed by historical global carbon-climate dynamics. Nonetheless, the diverse potential for global and regional carbon cycle dynamics to change sign under these scenarios highlights the continued need for improved comprehension of the major drivers of terrestrial carbon cycle dynamics.


The ocean carbon cycles of the models, and the thermal response of the climate system to greenhouse gas forcing, in general shows better qualitative agreement with each other (Figs. 1c, 6), but again ensemble spread increases after the 21st century in the high-emission scenario, and other surprises may be in store. In particular, two distinct types of lags in the physical system may lead to further warming beyond the time period in which greenhouse gases increase: if carbon emissions cease without

overshoot, lags in the physical climate may lead to continued warming after cessation, while in the overshoot case, mechanisms such as AMOC slowdown may temporarily obscure some of the warming, but then upon recovery of AMOC this temporary regional cooling may dissipate, leading to a resumption of warming long after the $CO_2$ has stabilized (fig. 10). If such dynamics

are real features of the Earth system, this would be of critical concern — even if we deploy large negative emissions, we still would have to have a plan for a world in which all they do is stabilise, rather than reduce, temperatures. Of particular note is that CESM2 showed a negative zero emissions commitment in MacDougall et al. (2020), despite showing the large 23rd-century warming with near-zero inferred emissions in the overshoot and stabilization scenario here, indicating that the ZEC framework as currently defined by MacDougall et al. (2020) as the temperature change evaluated 50 years following net-zero emissions may be insufficient for quantifying such lagged effects of $CO_2$ on climate. Further, we note that CESM2 shows the highest effective ECS of any of the models whose transient climate response is within the "likely range" as constrained by observed warming trends (Nijsse et al., 2020), because of this role of AMOC sensitivity acting to separate transient from equilibrium sensitivity (Hu et al., 2020). That the model satisfies the transient constraint underscores the possibility for nonlinearities in temperature versus cumulative emissions, although the long-term sensitivity may separately be constrained by paleoclimate evidence (Sanderson, 2020; Tierney et al., 2020).

Given that the plant-physiological and other $CO_2$ concentration-dependant processes represented in models are not routinely tested against observations from the highly out-of-sample conditions experienced under each of these scenarios (e.g. very high atmospheric $CO_2$ concentrations under SSP5-8.5 or rapidly decreasing $CO_2$ concentrations under SSP5-3.4-overshoot), it is to be expected that model differences will be large. Despite this, it is important to note that the ensemble spread in compatible emissions, which include all these uncertain carbon-cycle feedbacks, is relatively small when compared to the mean magnitude of the emissions themselves, particularly under the overshoot scenario (fig. 2a-b). Thus the uncertainty associated with carbon cycle feedbacks is relatively small compared to the anthropogenic emissions themselves. Further, the uncertainty does not conflict with the central result that warming is roughly proportional to cumulative emissions, with an additional temperature change for overshoot scenarios that is governed by the zero emissions commitment, at least for this century and the following one, and these results support the need for rapid reductions in $CO_2$ emissions to prevent the extreme impacts associated with warming.

Nonetheless, these results also suggest that there may be longer-term surprises in the coupled climate-carbon system to be encountered both in high-emissions and overshoot warming scenarios. The evident lack of consistent predictions in the terrestrial models, combined with the known structural differences, and the fact that none of the models include a complete set of processes that may be considered likely to affect the terrestrial carbon cycle, support the approach of accounting for feedbacks present in the Earth system but not included in ESMs, at least until greater convergence in terrestrial carbon cycle models can be shown. The ocean carbon cycle as well shows greater uncertainty on this time horizon than on the pre-2100 dynamics, for which there is greater agreement. The wide range of results shown here, despite a small number of models analyzed, also underscore the need for further testing of model dynamics on these longer timescales, the inclusion of more models and more systematic exploration of parameter and structural uncertainty on these longer-term dynamics, as well as the identification and use of observational constraints that are relevant to these longer-term dynamics of the coupled carbon and

climate systems. At the same time, unanticipated physical dynamics, such as the transient weakened-AMOC-driven cooling and its subsequent reversal may also be relevant on long timescales. Thus, we should continue to anticipate that surprises in the long timescale climate response are possible, even under relatively mitigated global warming scenarios, and which should
be taken into consideration when setting global climate policy.

## 5. Conclusions

We examine five CMIP6 ESMs, alongside a reduced-complexity EMIC, in a pair of experiments that extend to the year 2300, to explore the dynamics of the coupled carbon and climate systems on this timescale, which is longer than those typically considered in ESM analysis. We show that under contrasting high-emissions and overshoot scenarios warming is
approximately proportional to total cumulative $CO_2$ emissions, and for overshoot scenarios that deviations from this proportionality are primarily governed by the zero emissions commitment, but also that a further set of distinct deviations from linearity arise in some of the ESMs on post-2200 timescales. These multi-centennial deviations underscore the limits to our ability to coherently project the dynamics of the Earth system on these longer timescales. We note that, as on shorter timescales, the projections of terrestrial carbon dynamics differ most strongly between models, and that on the longer
timescales there is still enormous uncertainty in projected carbon dynamics. This uncertainty is evident in multiple ways: between the model projections of global carbon changes; between the model projections of the geographical regions contributing to feedbacks; between the pools responsible for the basic mechanisms of carbon cycle variability; and between one century to the next within models. We also show that lagged temperature effects leading to warming after cessation or reversal of emissions, beyond what has been shown in earlier or simpler models, may be possible outcomes in these projections.
These results show that a greater emphasis on identifying, attributing, and reducing uncertainty is needed on the wider range of possible futures that can be explored on these longer timescales, and that until such uncertainty can be reduced, we must anticipate and allow for surprises such as these in formulating global climate policy over these longer timescales.

## Data Availability

All CMIP6 data is available on the Earth system grid. IPSL-CM6A-LR output is available at: (Boucher et al., 2018, 2019b, a); CESM2-WACCM data available at: (Danabasoglu, 2019c, b, a); CanESM5 data available at: (Swart et al., 2019b, c, a); UKESM data available at: (Tang et al., 2019; Good et al., 2019a, b). UVic-ESCM output is available at (Mathesius and Zickfeld, 2021). CNRM-ESM2-1 data available at: (Seferian, 2018; Voldoire, 2019b, a). All analysis code is available as a jupyter notebook at https://github.com/ckoven/longterm_carboncycle/blob/master/longterm_carboncycle.ipynb (Koven,
560  2022)

## Author Contributions

C.K. and K.Z. conceived of study, with contributions from D.L.. All authors contributed to the design and/or use of one of the ESMs, EMIC, or long-term scenarios used in the analysis (CESM2-WACCM: C.K., R.F., D.L., K.L., M. Mills, B.S., W.W.; CanESM5: V.A., N.S.; IPSL-CM6A-LR: P.C.; UKESM1: C.J.; UVic-ESCM: S.M., K.Z.; CNRM-ESM2-1: R.S.; long-term scenarios: J.L., M. Meinshausen, Z.N.). C.K. performed data analysis and visualizations, with contributions from C.J., S.M., and K.Z.. C.K. wrote the initial manuscript draft. All authors provided input and feedback to manuscript text and figures, with particularly significant contributions from V.A., P.C., R.F., C.J., D.L., M. Meinshausen, Z.N., B.S., W.W., and K.Z..

## Competing Interests

The authors declare that they have no conflict of interest.

## Acknowledgements

C.K. was supported by the Director, Office of Science, Office of Biological and Environmental Research of the U.S. Department of Energy under Contract DE-AC02-05CH11231 through the Early Career Research Program, the Regional and Global Model Analysis Program (RUBISCO SFA), and the Next Generation Ecosystem Experiment-Tropics (NGEE-Tropics) project. R.F. was also supported by NGEE-Tropics. W.W., D.L. and R.F. were supported by the National Center for Atmospheric Research, which is sponsored by the U.S. National Science Foundation, which is a major facility sponsored by the National Science Foundation under cooperative agreement 1852977. Computing resources (doi:10.5065/D6RX99HX) were provided by the Climate Simulation Laboratory at NCAR's Computational and Information Systems Laboratory, sponsored by the National Science Foundation and other agencies. W.W. was supported by the U.S. Department of Agriculture NIFA award 2015-67003-23485 and NASA Interdisciplinary Science Program award number NNX17AK19G. B.S. and R.F are supported by H2020 programmes ESM2025 (grant agreement No 101003536) and 4C (GA 821003). C.D.J. was supported by the Joint UK BEIS/Defra Met Office Hadley Centre Climate Programme (GA01101) and the European Union's Horizon 2020 research and innovation programme CRESCENDO (grant agreement No 641816). R.S. acknowledges the European Union's Horizon 2020 research and innovation programme CRESCENDO (grant agreement No 641816) and ESM2025 – Earth System Models for the Future (grant agreement No 101003536). We acknowledge the World Climate Research Programme, which, through its Working Group on Coupled Modelling, coordinated and promoted CMIP6. We thank the climate modeling groups for producing and making available their model output, the Earth System Grid Federation (ESGF)

for archiving the data and providing access, and the multiple funding agencies who support CMIP6 and ESGF. K.Z. and S.M. acknowledge support from the Natural Science and Engineering Research Council of Canada's Discovery Grants Program.

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

**Figures**

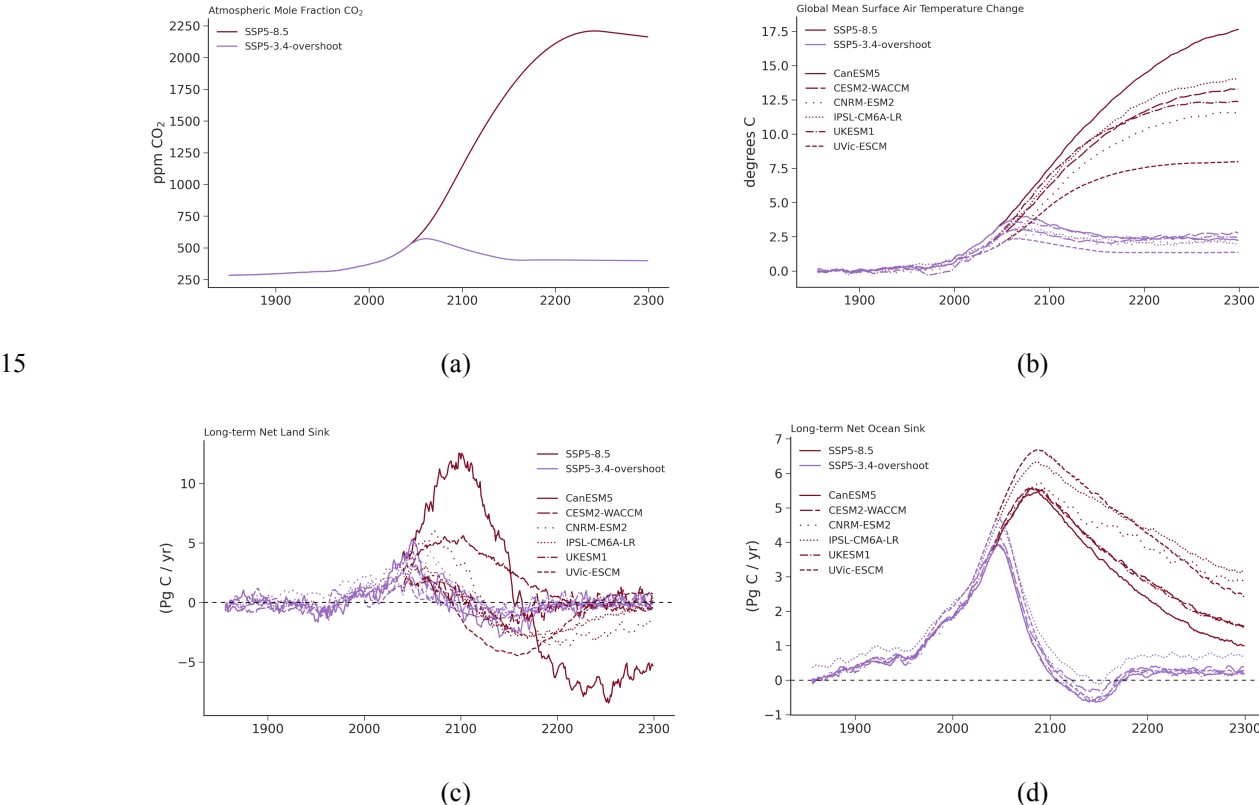

(a)                                     (b)

                         (c)                                    (d)

**Fig. 1 (a) Atmospheric CO$_2$ concentrations for the high emissions SSP5-8.5 and mitigated SSP5-3.4-overshoot scenario out to 2300. (b) Global-mean surface air temperature. (c-d) Long term dynamics, as projected by five ESMs and one EMIC, of (c) the terrestrial**
**carbon cycle, (d) the ocean carbon cycle, and, for both scenarios. All timeseries are smoothed to give 7-year running means, and positive flux represents a carbon sink into the land or ocean.**

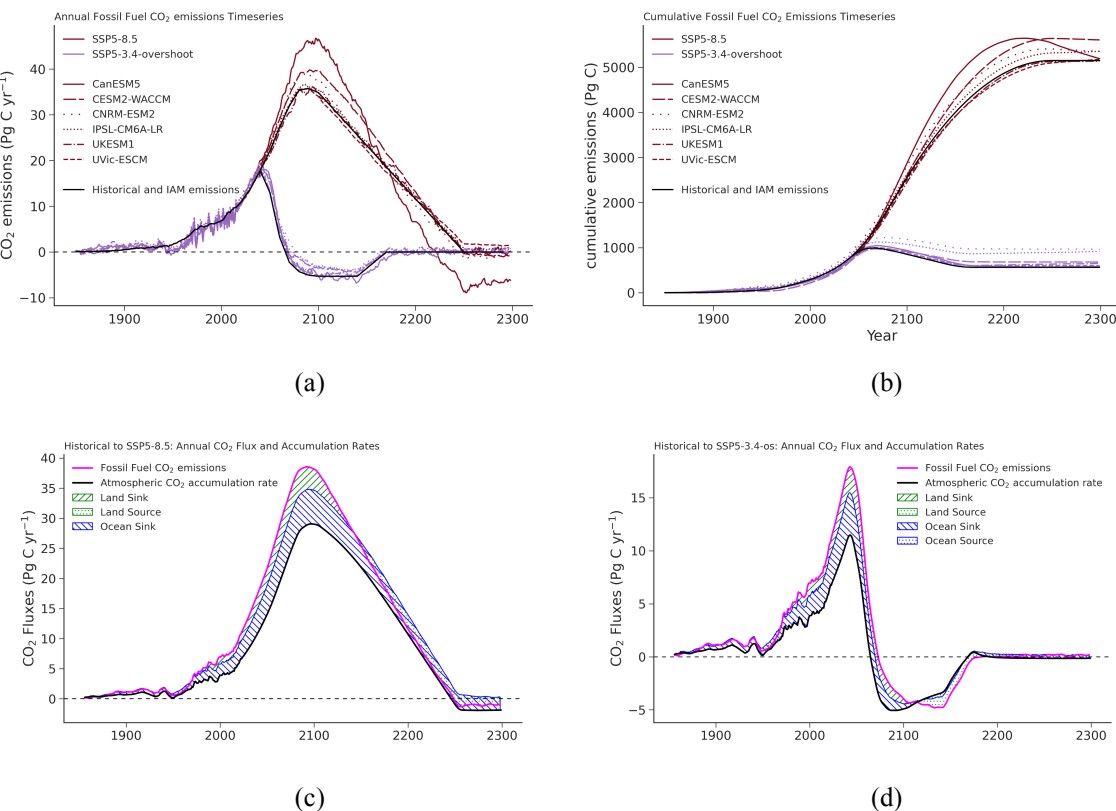

(a)

(b)

(c)

(d)

**Fig. 2 (a) ESM-inferred and IAM-specified harmonized annual fossil fuel (positive) and geologically sequesterred (negative) $CO_2$ emissions. (b) Cumulative fossil fuel (positive) and sequestration emissions (negative) as inferred by each ESM. (c-d) Ensemble-mean land, ocean, and fossil fuel emission fluxes shown together for the historical and future (c) SSP5-8.5 and (d) SSP5-3.4-overshoot scenarios. In (c-d), pink curves represent the annual fossil fuel $CO_2$ emissions, land and ocean sink fluxes are represented as hatched area and source fluxes are represented as stippled area, and the atmospheric $CO_2$ accumulation, which is the sum of fossil fuel, land, and ocean fluxes, is shown as the black curve.**

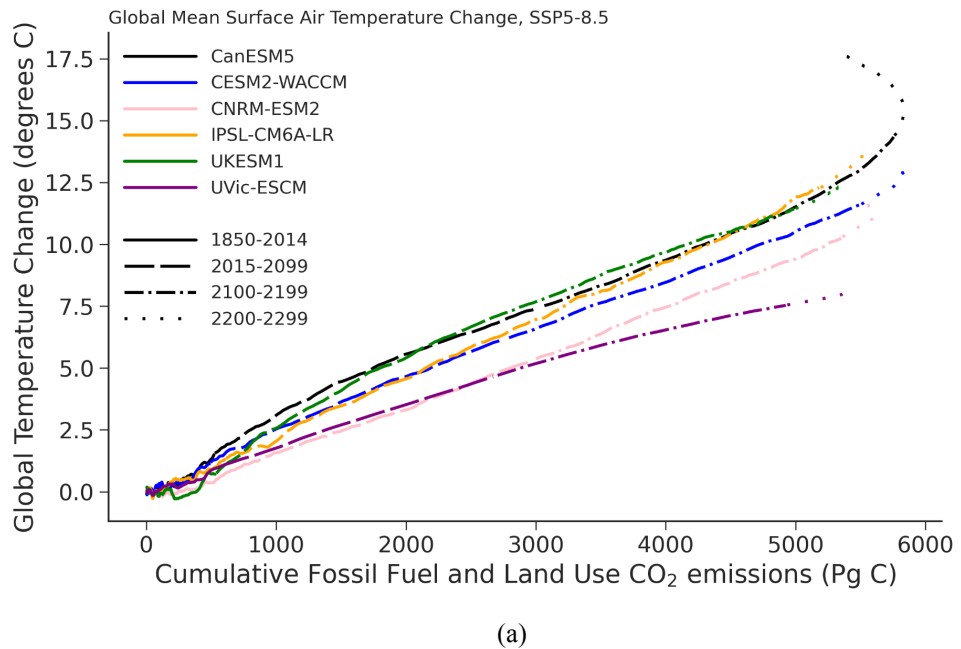

(a)

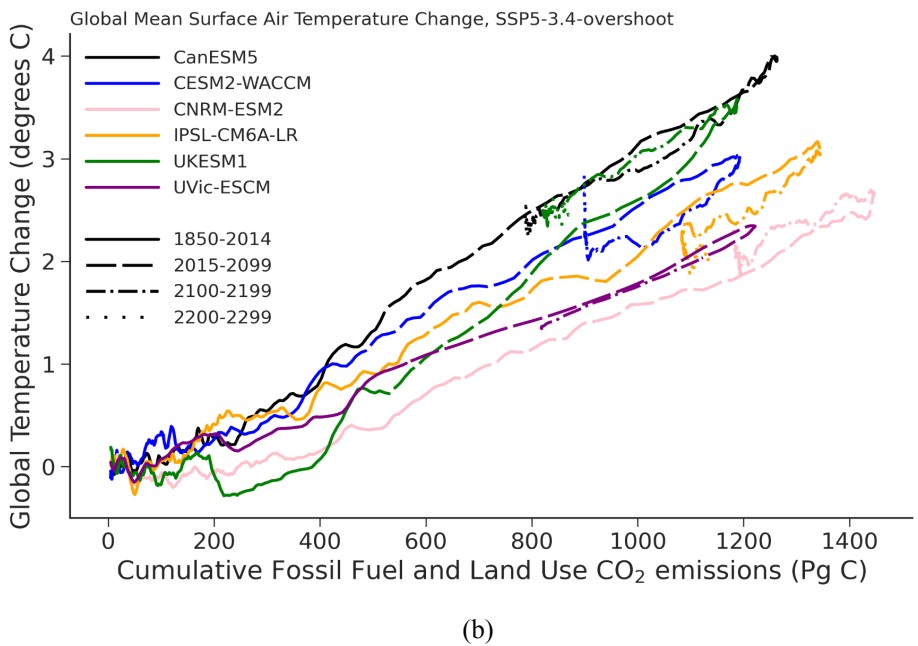

(b)

**940**    **Fig. 3 Global warming as a function of cumulative CO$_2$ emissions under the SSP5-8.5 (a) and SSP5-3.4-overshoot (b) scenarios. Emissions here are the sum of fossil fuel fluxes separately inferred for each ESM, and land use fluxes taken from the IAM that specified the two scenarios. Each model is identified by a color, and the time periods, broken into roughly centennial periods, are indicated by the dash patterns of the curves: historical (solid), 21st century (dash), 22nd century (dash-dot), 23rd century (dotted).**

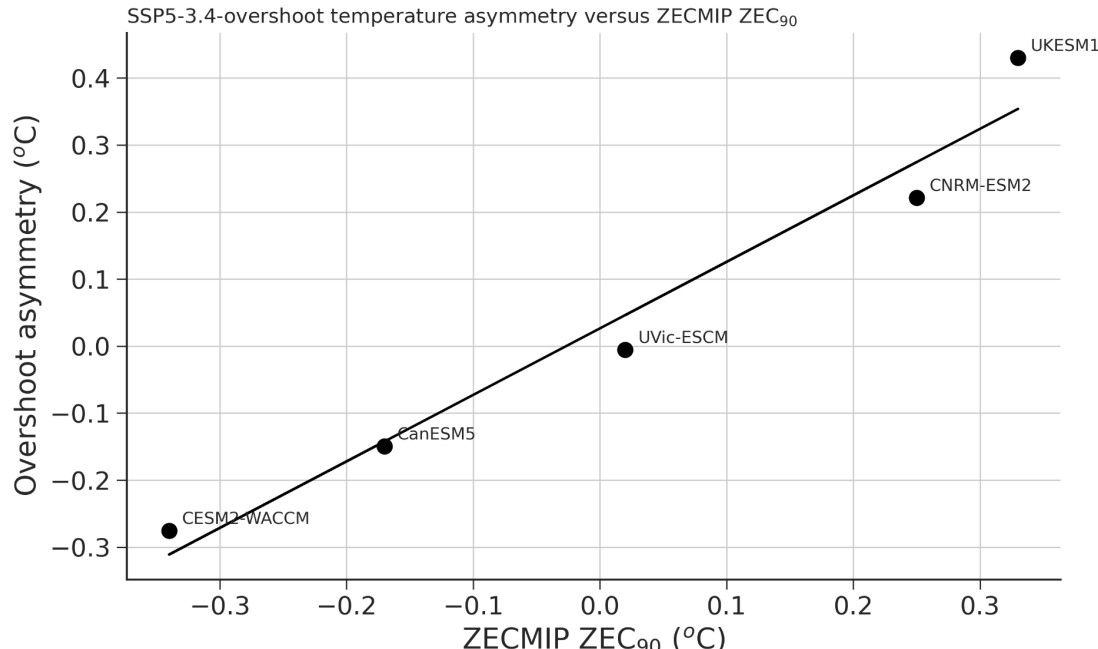

**Fig. 4. Comparison of the projected overshoot asymmetry in the temperature to cumulative emissions curve for each model for the SSP5-3.4-overshoot scenario against the zero emissions commitment (ZEC). Overshoot asymmetry is calculated as the temperature** 950 **difference at a given level of cumulative $CO_2$ emissions between the descending (negative $CO_2$ emissions) period and the ascending (positive $CO_2$ emissions) period. Here we evaluate this for the two 20-year periods centered at the point of 200 Pg C less than peak cumulative emissions for each model. ZEC values shown are the published values of 90-year zero emissions commitment ($ZEC_{90}$) from MacDougall et al. (2020).**

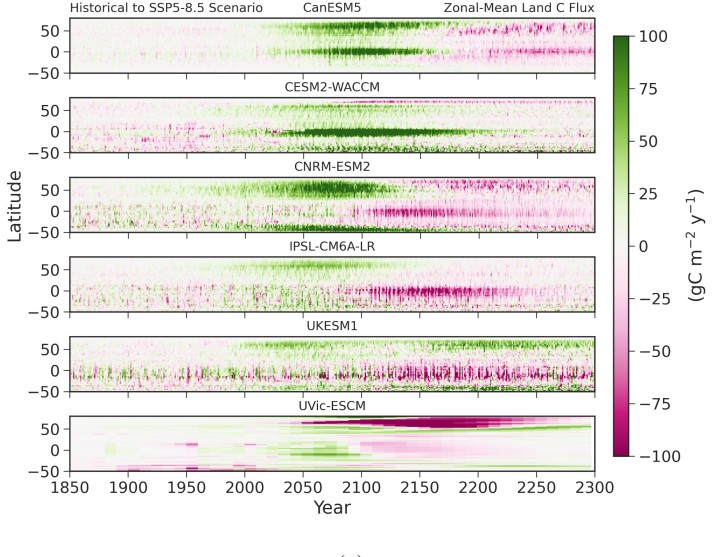

(a)

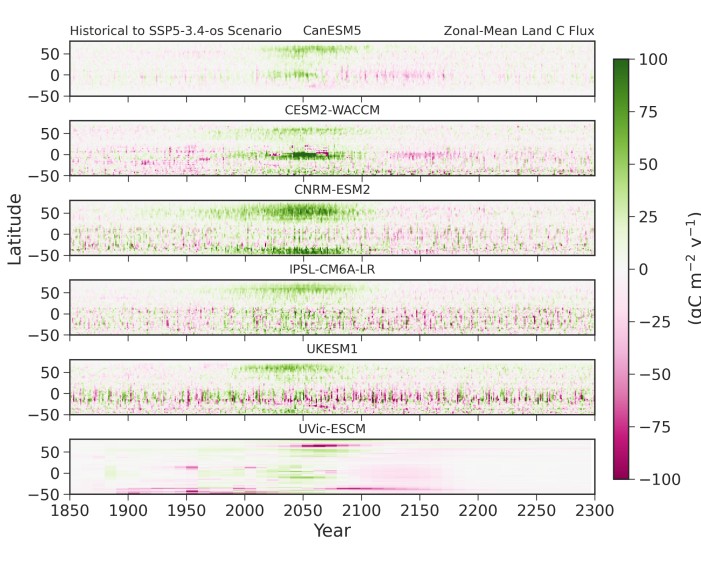

(b)

**Fig. 5 Zonal-mean terrestrial carbon flux dynamics of the six models under (a) the SSP5-8.5 and (b) SSP5-3.4-overshoot scenarios. Positive flux represents a net carbon sink.**

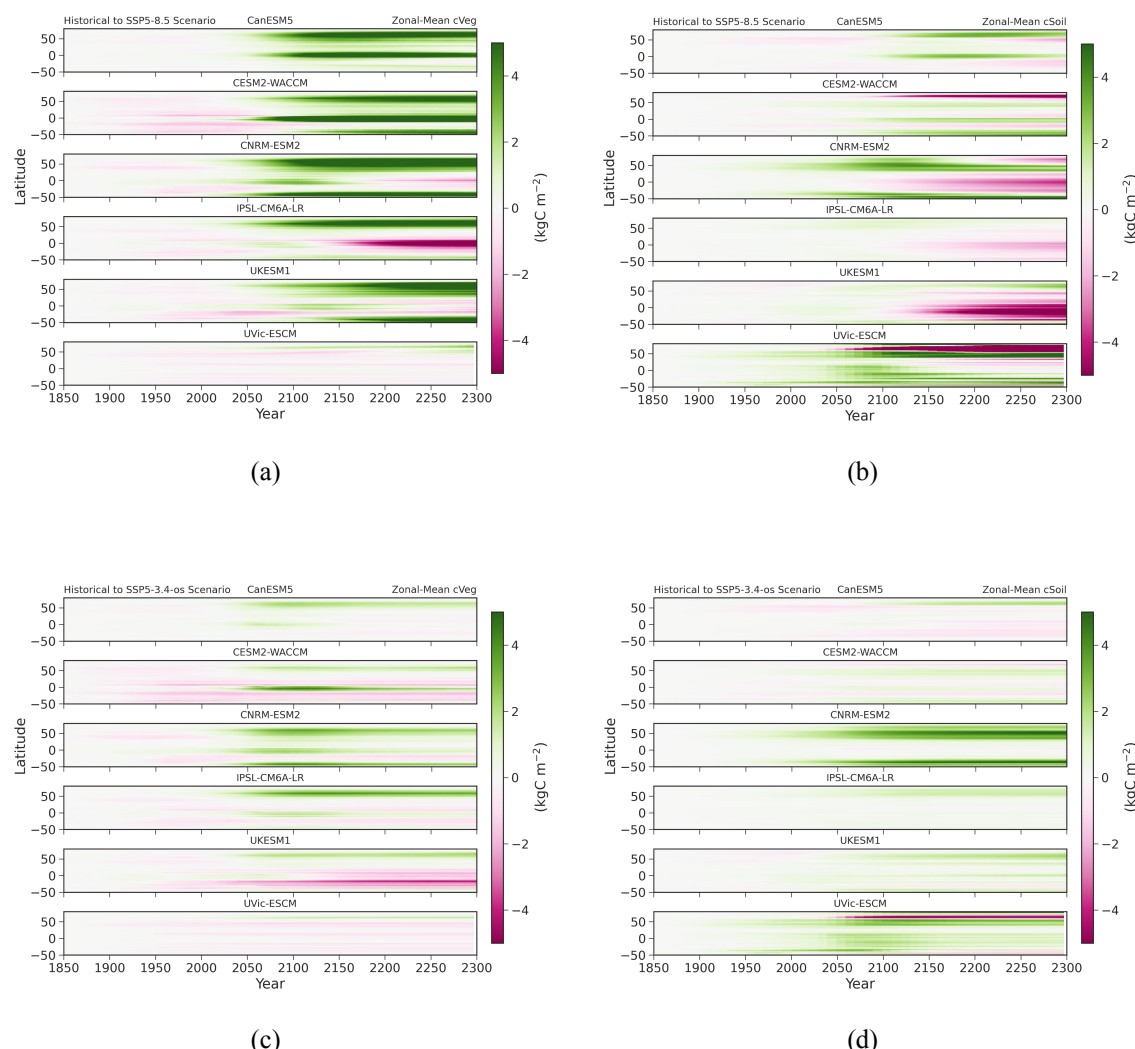

(a)

(b)


(c)

(d)

**Fig. 6 Zonal-mean changes to terrestrial vegetation (left column) and soil carbon stocks (right column) in the six models for SSP5-8.5 and SSP5-3.4os scenarios (top and bottom rows, respectively).**


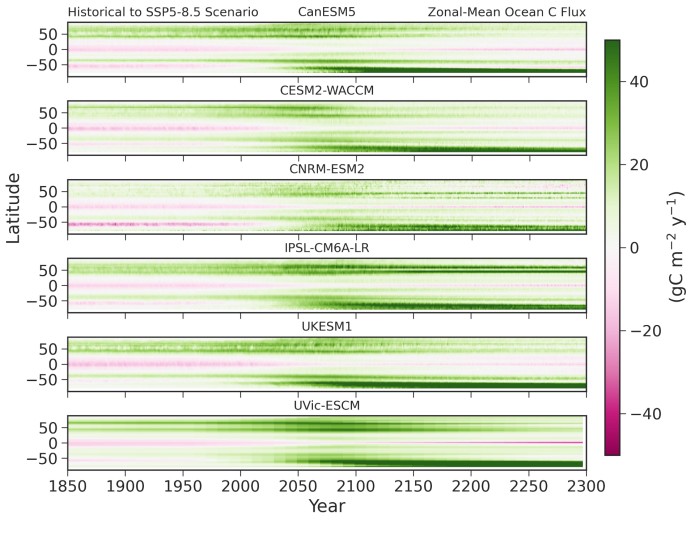

(a)

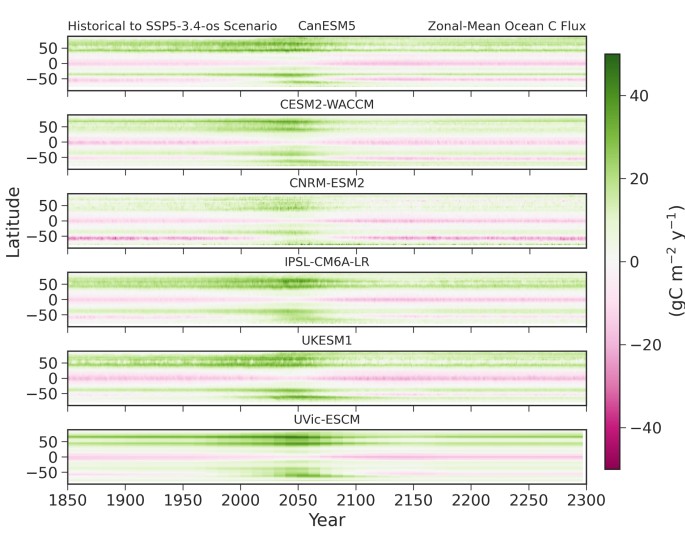

 (b)

**Fig. 7 Zonal-mean ocean carbon flux dynamics of the six models under (a) the SSP5-8.5 and (b) SSP5-3.4-overshoot scenarios. Positive flux represents a net carbon sink.**

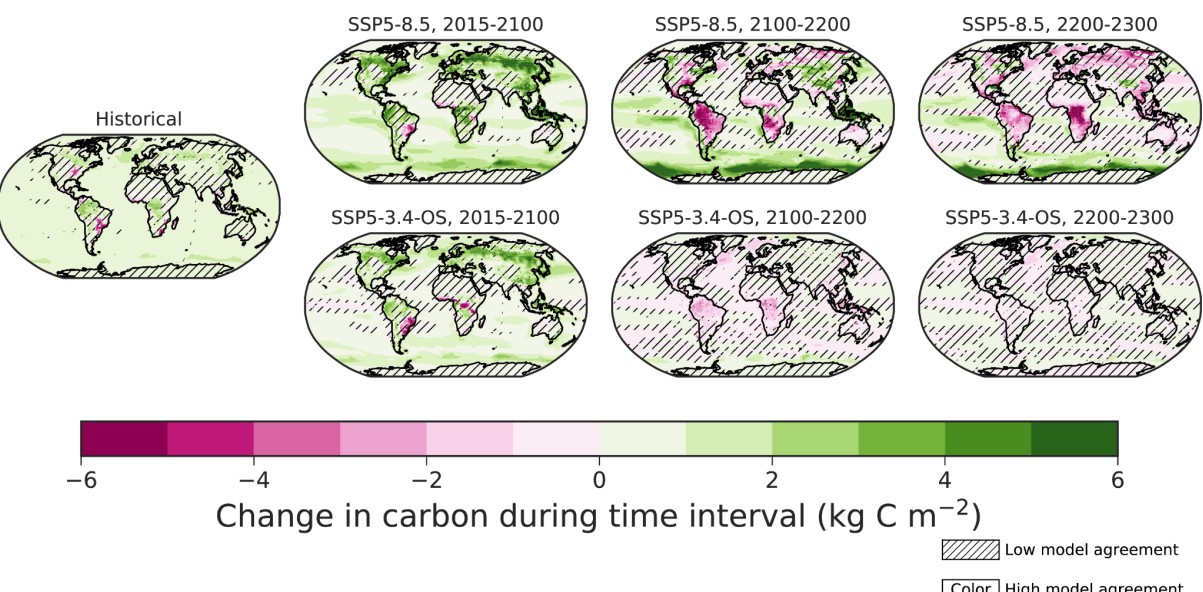

**Fig. 8 Maps of ensemble-mean projected carbon change for different scenarios and time periods. Carbon change is calculated as the time integral of carbon fluxes (on land) and time integral of flux anomalies relative to the first 20 years of historical simulation (on ocean). Hatching indicates that less than 83% (5 of 6) models agree with the ensemble mean on the sign of the carbon change. Positive flux represents a net carbon sink.**

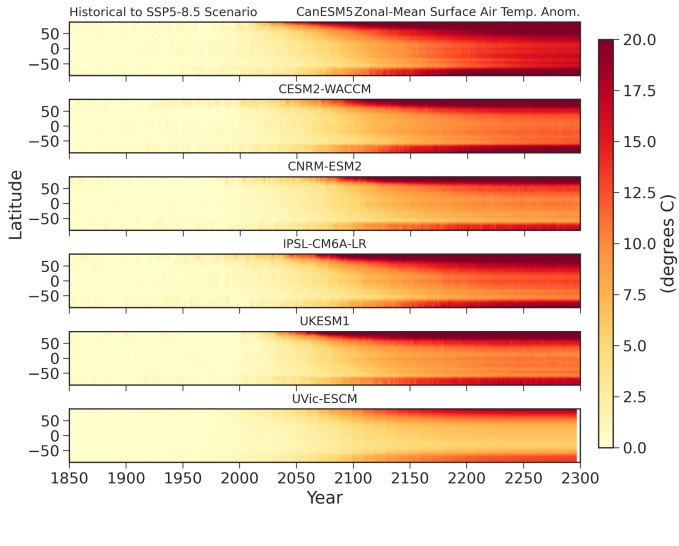

(a)

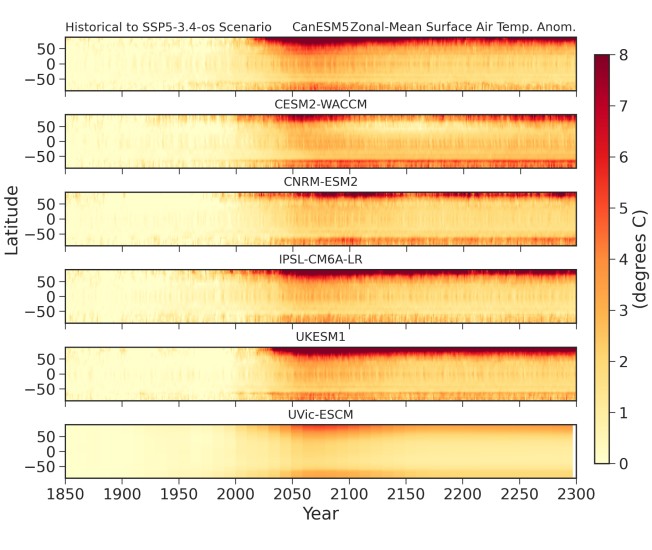

(b)

**Fig. 9 Zonal-mean temperature anomaly dynamics of the six models under (a) the SSP5-8.5 and (b) SSP5-3.4-overshoot scenarios. Note the different color scales for each panel.**

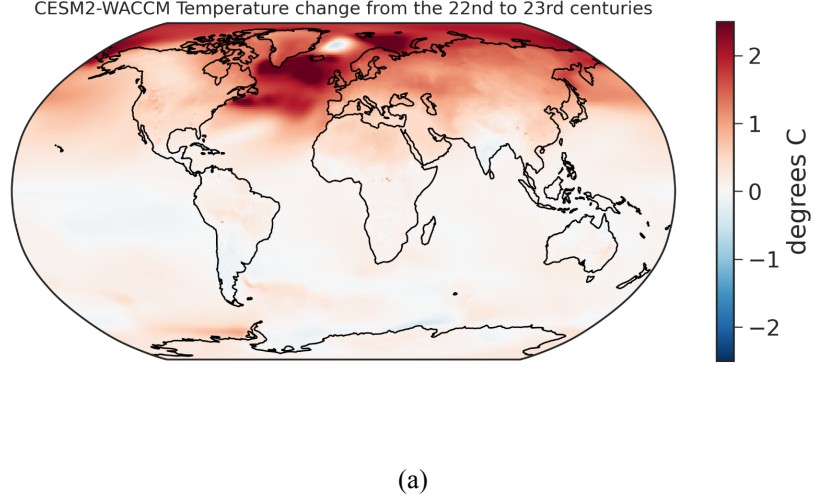

(a)

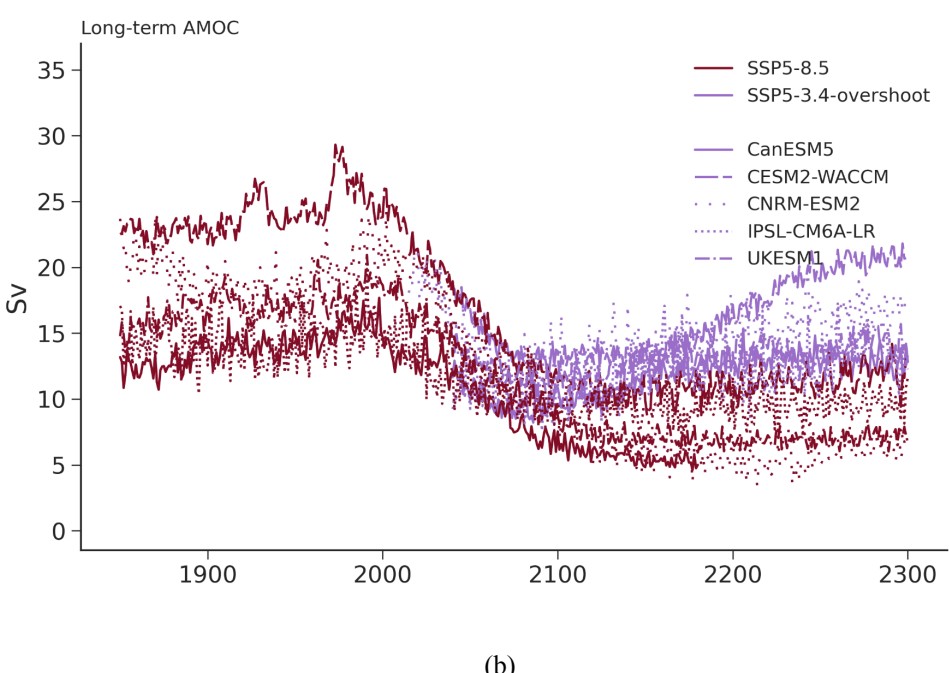


(b)

**Fig. 10 (a) Difference in the mean surface air temperature between the 22nd and 23rd centuries in the CESM2 model under the SSP5-3.4-overshoot. (b) Strength of the Atlantic Meridional Overturning Circulation for the all ESMs under both scenarios.**


**Appendix A**

Below we briefly describe relevant features of each of the models used in this study.


**A.1 CanESM5**

CanESM5 represents a major update since its predecessor CanESM2 (Arora et al., 2011), which was used in CMIP5, and is described in detail in Swart et al. (2019d). The resolution of CanESM5 (T63 or ~2.8° in the atmosphere and ~1° in the ocean) remains similar to CanESM2, and is at the lower end of the spectrum of CMIP6 models. CanAM5, the atmospheric component

of CanESM5, has several improvements relative to its predecessor including changes to clouds, aerosols, radiation, land surface, and lake processes.The land component in CanESM5 is represented using the Canadian Land Surface Scheme (CLASS) and the Canadian Terrestrial Ecosystem Model (CTEM) which simulate the physical and biogeochemical land surface processes, respectively. Together CLASS and CTEM calculate fluxes of water, energy, $CO_2$, and wetland $CH_4$ emissions at the land-atmosphere boundary. The introduction of dynamic wetlands and their (purely diagnostic) methane

emissions is a new biogeochemical process added since CanESM2. The nitrogen cycle over land is not represented but a parameterization of photosynthesis down-regulation as $CO_2$ increases is included. The physical ocean (OPA) and sea-ice (LIM2) components of CanESM5 are based on a customized version of NEMO version 3.4.1. The ocean is configured on the tripolar ORCA1 C-grid with 45 z-coordinate vertical levels, and a nominal horizontal resolution of 1°, with a refinement to 1/3° near the equator. The ocean carbon cycle is represented using the Canadian Model of Ocean Carbon (CMOC). The

biological component is a simple Nutrient-Phytoplankton-Zooplankton-Detritus (NPZD) model, with fixed Redfield stoichiometry, and simple parameterizations of iron limitation, nitrogen fixation, and export flux of calcium carbonate.

**A.2 CESM2-WACCM**

CESM2-WACCM is the whole atmosphere configuration of the CESM2 (Danabasoglu et al., 2020) model. This configuration,

which includes fully interactive stratospheric chemistry and dynamics, was used rather than the standard CESM2 configuration in order to more fully resolve the stratospheric response to the extreme warming in the SSP5-8.5 extension, as the standard CESM2 uses a set of atmospheric upper boundary conditions that are violated under the level of warming experienced in the long-term high-emissions scenario. The atmosphere is run at 0.9º x 1.25º resolution, and the ocean with a nominal 1º resolution. The model includes a full ocean model (Parallel Ocean Program version 2, POP2) with modularized biogeochemistry (Marine

Biogeochemistry Library, MARBL). The land model (Community Land Model, version 5, CLM5) is described in Lawrence

et al. (2019). Briefly, CLM5 includes a large number of changes and updates relative to the CLM4 version used in CESM1, including: a more detailed nitrogen cycle that allows for dynamic responses of N fixation, plant tissue stoichiometry, and leaf nitrogen allocation to changing nutrient limitations; a detailed crop model and more complete representation of land use; vertically-resolved soil biogeochemistry that includes permafrost carbon dynamics; acclimation of photosynthesis and plant respiration to changing temperature; and many others. Because of an artifact in model initialization procedure for soil carbon in CESM2, which left a set of grid cells in the High Arctic with unrealistically high values, here we apply a mask to exclude all the grid cells where vegetation productivity was equal to zero during a 100-year period of the preindustrial control simulation. This, alongside other model differences, including snow biases in the coupled model, also had the effect of reducing the permafrost carbon pool in CESM. Thus, while permafrost dynamics are permitted in CESM2 and CLM5, their feedback to warming is weaker than in the earlier CLM4.5 model as described in Koven et al. (2015).

## A.3 CNRM-ESM2-1

CNRM-ESM2-1 is the second generation Earth System model developed by CNRM-CERFACS for CMIP6 (Séférian et al., 2019). The atmosphere component of CNRM-ESM2-1 is based on version 6.3 of the global spectral model ARPEGE-Climat (ARPEGE-Climat_v6.3). ARPEGE-Climat resolves atmospheric dynamics and thermodynamics on a T127 triangular grid truncation that offers a spatial resolution of about 150 km in both longitude and latitude. CNRM-ESM2-1 employs a ''high-top'' configuration with 91 vertical levels that extend from the surface to 0.01 hPa in the mesosphere; 15 hybrid σ-pressure levels are available below 1500 m. The surface state variables and fluxes at the surface-atmosphere interface are simulated by the SURFEX modeling platform version 8.0 over the same grid and with the same time-step as the atmosphere model. SURFEXv8.0 encompasses several submodules for modeling the interactions between the atmosphere, the ocean, the lakes and the land surface.

Over the land surface, CNRM-ESM2-1 uses the ISBA-CTRIP land surface modeling system (Decharme et al., 2019; Delire et al., 2020) to solve energy, carbon and water budgets at the land surface. To simulate the land carbon cycle and vegetation-climate interactions, ISBA-CTRIP simulates plant physiology, carbon allocation and turnover, and carbon cycling through litter and soil. It includes a module for wildfires, land use and land cover changes, and carbon leaching through the soil and transport of dissolved organic carbon to the ocean. In the absence of nitrogen cycling within the vegetation, an implicit nitrogen limitation scheme that reduces specific leaf area with increasing $CO_2$ concentration was implemented in ISBA following the meta-analysis of (Yin, 2002). Additionally, there is an ad-hoc representation of photosynthesis down-regulation. During the decomposition process, some carbon is dissolved by water slowly percolating through the soil column. This dissolved organic carbon is transported by the rivers to the ocean. A detailed description of the terrestrial carbon cycle can be found in (Delire et al., 2020).

The ocean component of CNRM-ESM2-1 is the Nucleus for European Models of the Ocean (NEMO) version 3.6 (Madec et al., 2017), coupled to both the Global Experimental Leads and ice for ATmosphere and Ocean (GELATO) sea-ice model (Salas Mélia, 2002) version 6 and the marine biogeochemical model Pelagic Interaction Scheme for Carbon and Ecosystem Studies version 2-gas (PISCESv2-gas). NEMOv3.6 has a nominal resolution of 1° with a latitudinal grid refinement of 1/3° in the tropics. The ocean biogeochemical component of CNRM-ESM2-1 uses the Pelagic Interaction Scheme for Carbon and Ecosystem Studies model volume 2 version trace gases (PISCESv2-gas), which derives from PISCESv2 (Aumont et al., 2015). PISCESv2-gas simulates the distribution of five nutrients (from macronutrients: nitrate, ammonium, phosphate, and silicate to micronutrient: iron) which regulate the growth of two explicit phytoplankton classes (nanophytoplankton and diatoms). Dissolved inorganic carbon (DIC) and alkalinity (Alk) are involved in the computation of the carbonate chemistry, which is resolved by "Model the Ocean Carbonate SYstem" version 2 (MOCSY 2.0,Orr & Epitalon, (Orr and Epitalon, 2015)) in PISCESv2-gas. PISCESv2-gas uses several boundary conditions which represent the supply of nutrients from five different sources: atmospheric deposition, rivers, sediment mobilization, sea-ice and hydrothermal vents.

As shown in Séférian et al. (2019), CNRM-ESM2-1 does not simulate a net carbon balance close to zero. Modelling set-up of the ocean biogeochemical module was made to represent the mean pre-industrial ocean carbon outgassing consistently with the recent published estimates of Resplandy et al. (2018). The net imbalance in carbon fluxes is explained by the fact that PISCESv2-gas considers the riverine inputs of inorganic and organic carbon whereas in ISBA-CTRIP only represents the export of dissolved organic carbon. The export of dissolved inorganic carbon, particulate organic and inorganic carbon and calcium carbonate is assumed based on observed ratios between these species and DOC at river mouths. Because of the non-zero preindustrial carbon balance, and following (Liddicoat et al., 2021), we subtract the 500-year mean preindustrial land and ocean $CO_2$ fluxes from the transient historical and future fluxes in calculation of globally-integrated carbon fluxes.

**A.4 IPSL-CM6A-LR**

IPSL-CM6A-LR (Boucher et al., 2020) is the model which was used by the Institut Pierre-Simon Laplace (IPSL) to run most of the simulations needed in the context of the sixth phase of the Coupled Model Intercomparison Project. This coupled model includes the atmospheric LMDZ model, version 6A-LR (Hourdin et al., 2020), the ocean circulation NEMO model, version 3.6, (Madec et al., 2017), including sea ice NEMO-LIM3 model and thermodynamics and ocean biogeochemistry PISCES-v2 (Aumont et al., 2015), and the carbon cycle ORCHIDEE model, version 2.0 (Krinner et al., 2005). ORCHIDEE and PISCES models are coupled to the atmospheric LMDZ model via the OASIS3-MCT coupler (Marti et al., 2010). ORCHIDEE and LMDZ share the same spatial resolution of 2.5° x 1.3°, with the vertical atmospheric resolution being composed of 79 vertical levels up to 80km high. PISCES uses the eORCA1 quasi-isotropic global tripolar grid of 1°, with an additional refinement of 1/3° in the equatorial region and 75 levels in the vertical direction, with steps from 1 to 10m in the surface up to 200m at the bottom.

ORCHIDEE land surface model (version 2.0) does not include full nutrient cycles but does include a downregulation of maximum photosynthetic rates under high $CO_2$ concentrations. Based on Sellers et al. (1996), a logarithmic function was used for modeling the downregulation, using a reference $CO_2$ value of 380ppm. Moreover, the photosynthesis is calculated from radiation, soil moisture and temperature. The model includes fifteen PFTs that are grouped into three classes (tall vegetation, short vegetation, and bare ground) for the tiling of the land surface. These PFTs share the same leaf phenology but respond to different individual parameters. ORCHIDEE has an eleven-layer soil hydrology scheme, calculating its budget on a tile-basis to keep the balance in soil moisture distribution. Autotrophic and heterotrophic respiration is finally computed for different pools. Plant, litter and soil carbon pools are estimated on a modelled daily basis, compared to all other budgets, that are calculated every 15min, based on the atmospheric dynamics.

PISCES models various plankton types (phytoplankton, micro- and mesozooplankton) as well as the biogeochemical cycles of carbon, and main nutrients (P, N, Fe, and Si), where N, P and Si are the limiting nutrients in the phytoplankton's growth. The model has a fixed C:N:P ratio. Oceanic carbon and nutrients input into the model come from atmospheric deposition, river discharge in coastal regions and sediment transport.

**A.5 UKESM1**

UKESM1-0-LL is documented in Sellar et al (2019) and its configuration for CMIP6 simulations, including the ScenarioMIP runs is described in Sellar et al (2020). Land and atmosphere share the same horizontal grid: a regular latitude-longitude grid with $1.25º \times 1.875º$ resolution. There are 85 vertical levels extending to 85km in the stratosphere, and full stratosphere-troposphere atmospheric chemistry is simulated using the UKCA model. The ocean component uses the NEMO dynamical ocean on a nominally 1∘ tripolar grid with 75 vertical levels and an explicit nonlinear free surface.

The terrestrial biogeochemistry in UKESM1 is based on the land-surface model JULES (Clark et al., 2011; Best et al., 2011), but with some major enhancements developed for UKESM1. In particular the inclusion of a prognostic nitrogen cycle (Wiltshire et al., 2020) allows representation of limitations to carbon storage due to availability of nutrients. Parameters related to photosynthesis, respiration, and leaf turnover have been updated (Harper et al., 2016). The number of natural PFTs was increased from five to nine to represent the distinction between evergreen and deciduous plants and between tropical and temperate evergreen trees. The new dynamic vegetation and PFTs yield a closer match to observed vegetation distribution, with particular improvements to tropical and boreal forests and the high latitudes (Harper et al., 2018). The land use scheme designates a portion of each gridbox as cropland and a portion as pasture land, where only crops and pasture grasses can grow, respectively, to the exclusion of trees and shrubs. In the remainder of the gridbox, nine natural PFTs compete for space, which determines the distribution of forests, grasslands, shrublands, and bare soil.

Ocean biogeochemistry in UKESM1 is represented with the MEDUSA-2 model (The Model of Ecosystem Dynamics, nutrient Utilisation, Sequestration and Acidification; (Yool et al., 2013)): an intermediate-complexity plankton ecosystem model which resolves a dual size-structured ecosystem of small (nanophytoplankton and microzooplankton) and large (microphytoplankton and mesozooplankton) components. It explicitly includes the biogeochemical cycles of nitrogen, silicon, and iron nutrients as well as the cycles of carbon, alkalinity, and dissolved oxygen.

## A.6 UVic ESCM

The University of Victoria Earth System Climate Model (UVic ESCM) is a model of intermediate complexity with a horizontal grid resolution of $1.8°$(meridional) x $3.6°$(zonal). The version of the UVic ESCM used here (version 2.10) is described in detail in Mengis et al. (2020). UVic ESCM 2.10 includes a 3-D ocean general circulation model coupled to a dynamic-thermodynamic sea-ice model and a single layer energy-moisture balance model of the atmosphere with dynamical feedbacks. The land surface model is based on a simplified version of the Hadley Centre's MOSES land-surface scheme. New developments include a representation of soil freeze–thaw processes resolved in 14 subsurface layers (Avis et al., 2011), a multi-layer representation of soil carbon and soil respiration (MacDougall et al., 2012), and a representation of permafrost carbon, which is prognostically generated within the model using a diffusion-based scheme meant to approximate the process of cryoturbation (MacDougall and Knutti, 2016). In addition, the terrestrial component represents vegetation dynamics including five different plant functional types. The ocean carbon cycle is simulated by means of an OCMIP-type inorganic carbon-cycle model and a new marine ecosystem/biogeochemistry model solving prognostic equations for nutrients, phytoplankton, zooplankton and detritus (Keller et al., 2012). The new ocean biogeochemistry module includes phytoplankton light limitation, a more realistic zooplankton growth and grazing model, and an iron limitation scheme to constrain phytoplankton growth. Sediment processes are represented using an oxic-only calcium-carbonate model. Decadal-average values of spatially-explicit variables are used for this study.

**Appendix B**

| | CanESM5 | CESM2-WACCM | CNRM-ESM2-1 | IPSL-CM6A-LR | UKESM1 | UVic-ESCM | Ens. mean ± std dev |
|---|---|---|---|---|---|---|---|
| **Historical FF emissions 1850-2015** | 359 | 324 | 502 | 442 | 368 | 401 | 399 ±64 |
| **SSP5-8.5 FF emissions 2015-2100** | 2464 | 2234 | 2266 | 2148 | 2044 | 2074 | 2205 ±154 |
| **SSP5-8.5 FF emissions 2100-2200** | 2746 | 2749 | 2396 | 2404 | 2391 | 2263 | 2491 ±205 |
| **SSP5-8.5 FF emissions 2200-2300** | -375 | 301 | 194 | 361 | 331 | 437 | 208 ±296 |
| **SSP5-3.4-os FF emissions 2015-2100** | 584 | 592 | 684 | 633 | 496 | 496 | 581 ±75 |
| **SSP5-3.4-os FF emissions 2100-2200** | -364 | -235 | -212 | -202 | -250 | -296 | -260 ±61 |
| **SSP5-3.4-os FF emissions 2200-2300** | -1 | -1 | -10 | 44 | 37 | 2 | 12 ±23 |
| **Historical land sink 1850-2015** | -9 | -41 | 136 | 25 | -1 | 17 | 21 ±61 |
| **SSP5-8.5 land sink 2015-2100** | 550 | 305 | 342 | 180 | 123 | 85 | 264 ±172 |
| **SSP5-8.5 land sink 2100-2200** | 306 | 276 | -140 | -190 | -82 | -347 | -29 ±264 |
| **SSP5-8.5 land sink 2200-2300** | -650 | -32 | -255 | -127 | 3 | -8 | -178 ±251 |
| **SSP5-3.4-overshoot** | 172 | 162 | 257 | 170 | 79 | 36 | 146 ±78 |

| | | | | | | | |
|---|---|---|---|---|---|---|---|
| land sink 2015-2100 | | | | | | | |
| SSP5-3.4-overshoot land sink 2100-2200 | -143 | -41 | -11 | -41 | -36 | -88 | -60 ±48 |
| SSP5-3.4-overshoot land sink 2200-2300 | -8 | -20 | -20 | -13 | 29 | -13 | -7 ±18 |
| Historical ocean sink 1850-2015 | 123 | 120 | 120 | 173 | 125 | 139 | 133 ±20 |
| SSP5-8.5 ocean sink 2015-2100 | 365 | 380 | 376 | 420 | 372 | 441 | 392 ±31 |
| SSP5-8.5 ocean sink 2100-2200 | 364 | 397 | 459 | 518 | 397 | 535 | 445 ±71 |
| SSP5-8.5 ocean sink 2200-2300 | 155 | 213 | 328 | 368 | 208 | 324 | 266 ±85 |
| SSP5-3.4-overshoot ocean sink 2015-2100 | 203 | 222 | 219 | 254 | 208 | 251 | 226 ±22 |
| SSP5-3.4-overshoot ocean sink 2100-2200 | -23 | 4 | -3 | 37 | -16 | -10 | -2 ±21 |
| SSP5-3.4-overshoot ocean sink 2200-2300 | 20 | 32 | 23 | 70 | 21 | 28 | 32 ±19 |

1155 **Table B1. Cumulative fluxes by model, scenario, and time period. All fluxes are in Pg C.**

**Appendix C**

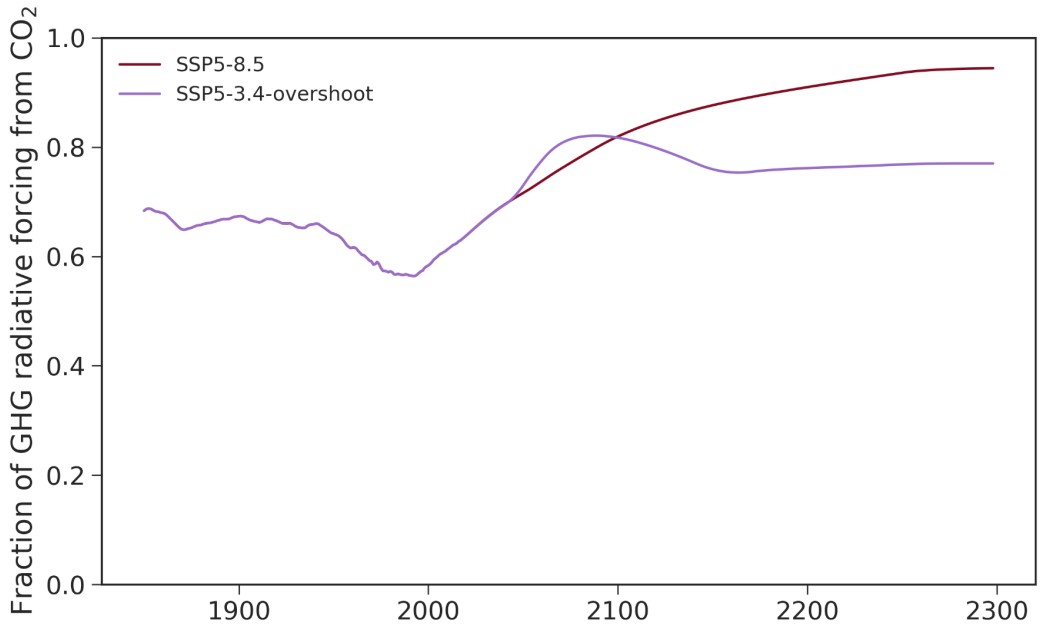

160 **Figure C1. Fraction of the total greenhouse gas radiative forcing from $CO_2$ for each of the scenarios, calculated using the specified concentrations of $CO_2$, $CH_4$, $N_2O$, CFC-12-eq, and HFC-134a-eq, with the radiative forcing calculations from Meinshausen et al., (2020). We show these approximate global-mean values, which will differ from the actual radiative forcing calculations within each model, because we do not have the diagnostics from each model to calculate their actual radiative forcing fractions.**