# Peer review of "Multi-century dynamics of the climate and carbon cycle under both high and net negative emissions scenarios"

_Earth System Dynamics, 2021_

## Referee Comment (RC2)

The manuscript by Koven et al. highlights results from long-term projections of the CMIP6 models. It addresses an important issue of a legacy of carbon emissions in the Earth System dynamics. The period until 2300 is long enough to establish responses of ocean and land uptakes to stabilized $CO_2$ concentrations. The paper is definitely important for the Earth Science community and should be published, but only after the authors address my concerns about three major weaknesses of the current manuscript.

1) First, the title promises surprises. Surprise is something one wouldn't expect, but here most of results are the same as one would expect from the MAGICC simulations or from simulations with ESMs until 2100. The last (very long) sentence in the abstract says about the possibility of surprises beyond the 21$^{st}$ century. Two reasons are: (i) the lack of agreement among the land models. I cannot count this as a surprise, it is a usual finding in the land model intercomparisons that land response to warming and $CO_2$ increase is very different among models, see e.g. Arora et al. (2020); (ii) a recovery and overshoot of AMOC above the pre-industrial level in CESM2-WACCM in SSP5-3.4 simulation. This feature is indeed interesting enough and can be counted as surprise, but in that case, I miss important details. What are the physical mechanisms – is it a feature of sea-ice instability or deep convection change? If this feature is brought to the top message in the title, the authors should invest more time into analysis of what have happened in the ocean circulation and carbon cycle response. Preferably they show a geographical map of patterns of land and ocean carbon uptake, either averaged over the last couple of decades or integrated over the 23$^{rd}$ century. The AMOC overshoot is a rare event with potentially important implications for the carbon cycle. Independently of how plausible the AMOC overshoot is, it makes sense to investigate how land/ocean carbon uptakes are changing in response to the AMOC recovery. Up to know, discussion in the literature was mostly about slowdown of AMOC and its effect on climate and carbon cycle. How does it work when AMOC overshoots, what are implication for ecosystems on land and carbon uptake in the ocean? The authors need either to make it clear and justify what was unexpected, or rename the paper and update the abstract.

2) Second, there are limitations of concentration-driven runs in analysis of TCRE (Fig. 3). Emissions in these runs are not purely anthropogenic but ESM-inferred emissions. Using monotonously increasing $CO_2$ scenarios is all right, but when concentrations start to decline, one can do wrong conclusions about temperature-cumulative emissions relationship if interactive carbon cycle is ignored. It is counter-intuitive: after cessation of fossil-fuel emissions in IAM scenarios, ocean naturally continues to take carbon, and ESM-inferring approach would count this ocean uptake as continuing anthropogenic emissions. Really confusing! This limitation must be discussed in depth for Fig. 3.

3) Third, about the paper conclusions. They are currently suboptimal and not that clear in terms of what new is found in these simulations. For example, a conclusion that land carbon uptake has many uncertainties – repeated several times - could be written without these simulations at all. This is very clear from ESM runs until 2100, and also can be seen in millennium-scale experiments, see eg Joos et al. (2013). Instead of these qualitatively vague conclusions on uncertainty, could you rather suggest how could we proceed in reducing uncertainty, what could be the main factors: response to droughts, $CO_2$ fertilization, natural vegetation dynamics, land use? I also would suggest to focus on what is common among the models, and not only on differences. This would be helpful for carbon cycle emulators to be used for analysis of other future scenarios.

Minor comments

P2., l.44: the long lifetime of $CO_2$ in the atmosphere: I miss here citations of papers focused on this issue, eg Archer et al. (2009), Joos et al. (2013) – please refer to them in the introduction.

Section 2.1: ScenarioMIP protocol doesn't define how landuse forcing (including wood and crop harvest) and aerosol forcing are implemented after 2100. Land cover forcing in Hurtt et al. (2020) is provided only until 2100. How is it extended beyond 2100? Are these forcing extensions treated in the same way in all models? These details should be explained.

l.114-115. This statement belongs to the Results section, not scenarios (unless scenario forcings were averaged).

Section 2.2 Model descriptions here is extensive and inconsistent among models. It doesn't allow to capture at the first glance a difference between models. Arora et al. (2020) did a better job with their Table 2 summarizing all important details of participating models. Some details like on the permafrost carbon dynamics in CESM could be reported in addition to the new Table.
Also: some descriptions here refer to components which are most likely not used in the study (such as wetland emissions in CanESM2). Why do you need to mention them?

p. 15, l. 467-472 – very long sentence, cut it in two after the reference. Interesting is the overshoot behavior.

P.15 – last sentence is unclear, what exactly is meant by reference to paleoclimate? Does it mean that evaluation by paleo runs allows to avoid surprises or other way around?

P16., l. 508 what is meant by budgeting here?

l. 512 How would you propose to test carbon model dynamics on long timescale, against what evidence? Emitting ca. 5000 PgC in RCP8.5 goes beyond a scale of any available evidence for the last 30 million years, and the quality of data for deep paleo is really poor.

Fig. 2 a-b: dashed lines are hard to see. Why don't you use the same color code for models as on figure 3? It will make figure consistent. Labels on c-d are hard to see on a paper, this figure is not readable on a paper.

Fig. 2: I suggest to add a table with values of emissions, land and ocean net fluxes at the years 2100, 2200, and 2300 for every model and for ensemble-mean. This would help to see a difference between models and time slices.

Fig. 2 caption: using the term "biospheric fluxes" for net land and ocean fluxes is confusing. Ocean carbon uptake is mostly inorganic, unless the authors could disentangle changes in anthropogenic uptakes due to solubility and biology. I suggest consistently use land and ocean uptakes. Strictly speaking, landuse emissions are anthropogenic too, so one rather has to use "fossil-fuel" emission term.

Fig. 3. There is a principal inconsistency between C-driven and E-driven TCRE plots, as explained in the second major comment. This has implications for this plot, as C dynamics of MAGICC are different from the ones of ESMs. A negative betta feedback in E-driven run will likely stabilize the system much faster than in the C-driven simulation. Please discuss it.

Fig. 4-5. These Hovmöller diagrams are very useful, but they hide the fact that the zonal land distribution is inequal. This is especially valid for the latitudes to the south of 40°S. I suggest to cutoff these diagrams at 40°S or 50°S.
Also - ESMs have 2-D geographic distribution of carbon. You can add a figure of ensemble mean for land and ocean net fluxes at 2300 for two scenarios. It will be very useful to see such a plot corresponding to Fig. 2 c-d.

References

Archer, D., Eby, M., Brovkin, V., Ridgwell, A., Cao, L., Mikolajewicz, U., Caldeira, K., Matsumoto, K., Munhoven, G., Montenegro, A., Tokos, K., 2009. Atmospheric lifetime of fossil-fuel carbon dioxide, Annual Reviews of Earth and Planetary Sciences, 37, 117-134.

Joos, F., Roth, R., Fuglestvedt, J. S., Peters, G. P., Enting, I. G., von Bloh, W., Brovkin, V., Burke, E. J., Eby, M., Edwards, N. R., Friedrich, T., Frölicher, T. L., Halloran, P. R., Holden, P. B., Jones, C., Kleinen, T., Mackenzie, F. T., Matsumoto, K., Meinshausen, M., Plattner, G.-K., Reisinger, A., Segschneider, J., Shaffer, G., Steinacher, M., Strassmann, K., Tanaka, K., Timmermann, A., and Weaver, A. J.: Carbon dioxide and climate impulse response functions for the computation of greenhouse gas metrics: a multi-model analysis, Atmos. Chem. Phys., 13, 2793–2825, https://doi.org/10.5194/acp-13-2793-2013, 2013.

---

## Author Response (AR1)

We thank both referees as well as the editor for their time and care in handling this manuscript. In this point-by-point response, our original responses to the referees have been left in blue text, and the new responses with specific changes are in green.

Reviewer 1 (Jörg Schwinger) Comments:

The authors present an analysis of the CMIP6 ScenarioMIP SSP5-8.5 and SSP5-3.4-OS scenarios and their extension to 2300, which have been simulated by 4 ESMs and one EMIC. This pair of scenarios initially follows the same pathway, but diverges after 2040. SSP5-8.5 represents an extreme case of increasing fossil fuel use, leading to very high atmospheric $CO_2$ concentrations, while SSP5-3.4-OS assumes aggressive climate mitigation after 2040 including a large amount of net negative $CO_2$ emissions. The authors focus on the long term (beyond the 21st century) climate and carbon cycle response in these contrasting scenarios. Carbon fluxes between the land surface and the ocean, the transition of these fluxes between source and sink, the surface temperature response, and the proportionality between warming and cumulative carbon emissions are investigated. The main findings include a very large model uncertainty in the land components of the ESMs, which is particularly pronounced for the high emission scenario.

This manuscript is an important contribution to the analysis of the wealth of CMIP6 model data, and the first one (to my knowledge) describing results from the SSP-extensions. It fits well into the scope of ESD and will be of interest to a broad readership. The manuscript is generally well written and well structured. I did not find any serious problems with the manuscript, and I recommend it for publication in ESD after a few rather minor comments and suggestions have been considered by the authors.

We thank the reviewer for their positive assessment of the manuscript.

Main points:

1) The treatment of land-use/land-use change emission in this analysis could be explained better, particularly in the context of negative emissions and BECCS. The negative emissions in the SSP5-3.4-OS are generated (mainly?) through BECCS, I assume, at least there is a massive expansion of cropland in the overshoot scenario (more than a doubling between ~2050 to ~2070 at the expense of pasture, O'Neill et al. 2016, Fig. 4). On the other hand, the ESMs employed here will not represent anything like BECCS. At most, I guess, they will represent harvesting of crops (but release this carbon back to the atmosphere)? There is only limited information on this issue in the manuscript. The sentence in line 105-107 is unclear to

me, particularly the part "thus can be separately inferred..." - but this inference is not done in this work? It would be useful to know how (if?) the massively increased crop area in the overshoot scenario is treated in the models, and what this implies for the analysis.

Related to this, in section 3.3 the authors diagnose fossil fuel and industrial emissions and show that these compare reasonably well with the IAM emissions. This must mean that the carbon uptake by bioenergy crops for BECCS in the IAM is counted as "fossil fuel and industrial"? Also, in Fig. 4b, the large and quite abrupt change in cropland area in the overshoot scenario seems to entail no land-use change emission flux (land-use change emissions are part of the depicted flux, are they?). Is this because a transformation of pasture to cropland has no large effect? This would probably be different if the cropland expansion would happen on the cost of forest?

It would be nice if some of these aspects of negative emissions and how ESMs represent or do not represent them could be covered in the methods section (in the existing subsections or maybe add an extra subsection).

This is a very good point and we will add further discussion of this in the manuscript. In particular, we will plan to include information in the description of each model for (a) whether they treat pasture and cropland as distinct land-use categories, (b) how they handle the transition between pasture and cropland, and (c) if and how carbon fluxes associated with BECCS are treated differently from carbon associated with non-BECCS crops.

The reviewer was correct to point out that the the sentence from original lines 105-017 that had read "thus can be separately inferred" was in fact the opposite of our intended meaning, and have changed it to now read, "Land-use-driven carbon emissions are directly reflected in changes in the terrestrial carbon inventories and thus cannot be separately inferred based on terrestrial model dynamics themselves, as they are mixed with the model responses to changing climate and $CO_2$"

We have also now added the following sentence to the text to address the comment about BECCS: "These net negative emissions are largely driven by biomass energy with carbon capture and sequestration (BECCS), which in the land-use drivers of the scenarios is associated with a large conversion of pasture to crop lands (O'Neill et al. 2016), however none of the models explicitly track BECCS-related harvest fluxes."

We also include a sentence detailing which of the models differentiate between pasture and crop land-use types: "Three of the models here (CESM2-WACCM, CNRM-ESM2-1, and UKESM1) distinguish between crop and pasture lands, which is relevant to the overshoot scenario and its large expansion of croplands from pasture."

In addition, we now also do include the harmonized land-use emission fluxes that were used to generate the $CO_2$ concentrations in addition to the ESM-compatible fossil fuel emissions in figure 3 that shows temperature versus cumulative $CO_2$ emissions.

2) The discussion of the role of non-$CO_2$ forcings in the TCRE relationship could be a bit more comprehensive. It would be interesting to know the long-term development of non-$CO_2$ forcings as specified in the scenarios (i.e., fixed at 2100 levels or linearly declining, etc.). Currently it is only mentioned that the fraction of non-$CO_2$ forcing is larger in the early parts of the scenarios (lines 310-311), maybe the authors can expand this a bit.

We will add further discussion of this point in the manuscript.  The non-$CO_2$ timeseries for these long-term scenarios can be seen in figure 2 (emissions), figure 7 (radiative forcing), and figure 8 (concentrations) of Meinshausen et al. (2020, https://doi.org/10.5194/gmd-13-3571-2020)

We have now added a figure C1 in Appendix C, which shows the CO2 radiative forcing as a fraction of the total greenhouse gas radiative forcing, and refer to this in the text.  The absolute values and their radiative forcing can be found in Meinshausen et al. (2020).

3) The manuscript would benefit from moving the individual model descriptions to an Appendix. Instead, in the methods section, only a summary could be presented, but this should contain key features like the inclusion of nitrogen cycle, dynamic vegetation, and soil physics (which could be moved from the Discussion-section lines 459-464). Also, this would be a place to say something about how land use change is handled in the models. This is a personal preference, but I think this way it would be easier for the reader to grasp the most relevant differences between the models.

This is a good idea and we are happy to make such a change in the manuscript.

We have now moved all model description text to the appendix A1.

4) Finally, I find the "23rd Century surprises" in the title is pushing it a bit. The authors argue that the "lack of agreement among land models on the mechanisms and geographic patterns of carbon cycle feedbacks, alongside the potential for lagged physical climate dynamics to cause warming long after $CO_2$ concentrations have stabilized, point to the possibility of surprises in the climate system beyond the 21st century time horizon, even under relatively mitigated global warming scenarios" (abstract lines 36-40). First, I wouldn't agree that the lack of agreement among land models is comparable between the very high and the mitigated scenario - Fig. 1c shows that on the global scale land models disagree much more for the strong forcing (also for Fig. 4b I would argue that the models do show some similarity for the mitigated scenario). Second, the fact that land models disagree widely does not mean that there will be or could be surprises - as long as the model results bracket the real-world behavior (which, of course, we cannot know, but this is a fundamental problem of all our science). It is a personal preference, but I would simply delete "23rd Century surprises" from the title and reword lines 36-40 in the abstract (and the other occurrences where the authors argue for "other surprises being in store" or similar).

As both reviewers agree that the current title is not working well for the manuscript, we will change it in revisions, most likely to something like "Multi-century dynamics of the climate and carbon cycle under both high and net negative emissions scenarios".

We have now changed the title to "Multi-century dynamics of the climate and carbon cycle under both high and net negative emissions scenarios".

Minor points:

lines 26-27: "...followed by stabilization of atmospheric $CO_2$ concentrations by means of large net-negative $CO_2$ emissions." This is not very precise: the large net-negative $CO_2$ emissions do not stabilize atmospheric $CO_2$ but decrease it.

We will reword this in revisions.

We have now reworded this to say "followed by a decrease of atmospheric CO2 concentrations".

line 30: "climate-carbon feedbacks" - I suggest replacing this by the more general term "carbon-cycle feedbacks" (for the ocean a large part of the weakening comes from reduced buffer capacity of the upper ocean, which (in this terminology) is a "concentration-carbon feedback"

We will reword this in revisions.

We have now reworded this to say "albeit weakened by carbon cycle feedbacks".

line 232-234: "This implies a substantial slow component in the models which continue to warm past the period of $CO_2$ stabilization, beyond the effective transient values reported above..." The TCR values reported above apply at 2xCO$_2$ in the 1% scenario, I do not really understand the connection here. Whether or not one would expect continued warming depends on remaining (implied) $CO_2$ emissions, non-$CO_2$ forcing and the ZEC of those models. Could the authors please clarify this?

We will add more detail on this point in revisions.

We have reworded this to clarify that it is the physical response that we are talking about, and not the coupled physical-carbon cycle. This sentence now reads: "Since these are concentration-forced experiments, this divergence in long-term warming after stabilization of $CO_2$ concentration implies a substantial slow component to the physical climate feedback in the models that continue to warm, beyond the effective transient values reported above, which reflect short-to-medium term feedback processes that dominate the TCR (and implicitly the TCRE) (Proistosescu and Huybers 2017)"

Technical:

We will make each of these editorial changes below in revisions.

line 25: I suggest deleting "second"

Done.

line 76-77: Please check and revise this sentence, it does not make sense to me.

Thanks for catching, this sentence was unfinished in the prior version. It now reads "Further, because these models all report more detailed information that can allow some degree of process attribution to the dynamics, we separate the carbon cycle responses geographically, land from ocean, and on land we separate the soil and vegetation responses"

line 102: "...global-mean timeseries, but do not feed back on atmospheric $CO_2$ (Fig. 1a)." -> "...global-mean timeseries (Fig. 1a), but do not feed back on atmospheric $CO_2$."

Done.

line 223: use the degree-sign instead of spelling out "degree"

Done.

line 259: "..., and consistent ..." -> "..., and which is consistent..." (but this is also a very long sentence, consider splitting in two)

Thanks. We have split this into two sentences: "For the SSP5-3.4-overshoot scenario, model agreement of the terrestrial carbon cycle is much higher, with all models transitioning from sink to source during the late 21st or early 22nd centuries, which counteracts some of the net-negative anthropogenic emissions by that time in terms of their effect on lowering atmospheric $CO_2$ concentrations. This change in sign is consistent with the CMIP5 RCP2.6 results shown in Jones et al. (2016)."

line 347: "near-future time period" - please be more specific, up to 2040 or 2050?

Added a parenthetical "(prior to 2040)" here.

line 435: "to emissions" -> "to cumulative emissions"

Done.

line 490: "unless behavior can also be constrained" - please check the grammar of this sentence (maybe consider splitting in two)

Thanks, we have split this sentence in two and added more detail. It now reads "Further, we note that CESM2 shows the highest effective ECS of any of the models whose transient climate response is within the "likely range" as constrained by observed warming trends (Nijsse, Cox, and Williamson 2020), because of this role of AMOC sensitivity acting to separate transient from equilibrium sensitivity (Hu et al. 2020). That the model satisfies the transient constraint underscores the possibility for nonlinearities in temperature versus cumulative emissions unless long-term sensitivity can also be constrained by paleoclimate evidence (Sanderson 2020; Tierney et al. 2020)."

Reviewer 2 (Victor Brovkin) Comments:

The manuscript by Koven et al. highlights results from long-term projections of the CMIP6 models. It addresses an important issue of a legacy of carbon emissions in the Earth System dynamics. The period until 2300 is long enough to establish responses of ocean and land uptakes to stabilized $CO_2$ concentrations. The paper is definitely important for the Earth Science community and should be published, but only after the authors address my concerns about three major weaknesses of the current manuscript.

We thank the reviewer for their positive assessment of the manuscript.

1) First, the title promises surprises. Surprise is something one wouldn't expect, but here most of results are the same as one would expect from the MAGICC simulations or from simulations with ESMs until 2100. The last (very long) sentence in the abstract says about the possibility of surprises beyond the 21st century. Two reasons are: (i) the lack of agreement among the land models. I cannot count this as a surprise, it is a usual finding in the land model intercomparisons that land response to warming and $CO_2$ increase is very different among models, see e.g. Arora et al. (2020); (ii) a recovery and overshoot of AMOC above the pre-industrial level in CESM2-WACCM in SSP5-3.4 simulation. This feature is indeed interesting enough and can be counted as surprise, but in that case, I miss important details. What are the physical mechanisms – is it a feature of sea-ice instability or deep convection change? If this feature is brought to the top message in the title, the authors should invest more time into analysis of what have happened in the ocean circulation and carbon cycle response. Preferably they show a geographical map of patterns of land and ocean carbon uptake, either averaged over the last couple of decades or integrated over the 23rd century. The AMOC overshoot is a rare event with potentially important implications for the carbon cycle. Independently of how plausible the AMOC overshoot is, it makes sense to investigate how land/ocean carbon uptakes are changing in response to the AMOC recovery. Up to know, discussion in the literature was mostly about slowdown of AMOC and its effect on climate and carbon cycle. How does it work when AMOC overshoots, what are implication for ecosystems on land and carbon uptake in the ocean? The authors need either to make it clear and justify what was unexpected, or rename the paper and update the abstract.

As both reviewers agree that the current title is not working well for the manuscript, we will change it in revisions, most likely to something like "Multi-century dynamics of the climate and carbon cycle under both high and net negative emissions scenarios". The degree of surprisingness of these dynamics is of course subjective, but we can indeed focus a bit more on the AMOC dynamics. What is interesting about the AMOC recovery on the global climate is not that it drives a strong carbon cycle response (CESM2 has essentially a constant ocean sink after about 2170 in the SSP5-3.4-overshoot scenario), but rather that it appears to drive a

physical warming response.  We can add more detail on these dynamics in the discussion of that result.

We have now changed the title to "Multi-century dynamics of the climate and carbon cycle under both high and net negative emissions scenarios". We have also made several of the recommendation shere, including maps of the geographical patterns of uptake, and more detail (including a table) on the changing magnitude of the model ensemble spread and how it reflects changes to the uncertainty of projections. We have decided not to go into greater detail on the AMOC recovery dynamics than in the prior draft, since that dynamic is specific to one model and will benefit from a more detailed treatment and comparison with long-term behavior in other scenarios in another paper.

2)  Second, there are limitations of concentration-driven runs in analysis of TCRE (Fig. 3). Emissions in these runs are not purely anthropogenic but ESM-inferred emissions. Using monotonously increasing $CO_2$ scenarios is all right, but when concentrations start to decline, one can do wrong conclusions about temperature-cumulative emissions relationship if interactive carbon cycle is ignored. It is counter-intuitive: after cessation of fossil-fuel emissions in IAM scenarios, ocean naturally continues to take carbon, and ESM-inferring approach would count this ocean uptake as continuing anthropogenic emissions. Really confusing! This limitation must be discussed in depth for Fig. 3.

We will add further discussion of this in revision.  However, the point that the reviewer makes here isn't necessarily true, so long as the emulator used by the IAM that prescribes the concentration trajectory for a given scenario is doing a good job of emulating the Earth system. For example, in the zero-emissions condition that the reviewer refers to, the atmospheric $CO_2$ concentration will decline in accordance with the continued ocean carbon uptake.  Because ESM-inferred emissions are calculated based on the change in time in the integr ated carbon summed across the land, ocean, and atmosphere together, an atmosphere-to-ocean carbon flux should not lead to any inferred anthropogenic emissions at all.  If the IAM and ESM were to disagree on the relative amount of ocean and land uptake in response to the forcings, then the atmospheric $CO_2$ concentration would not reflect the ESM fluxes, and the inferred emissions from the ESMs would deviate from those generated by the IAM and used to construct the scenario, and this is why we compare these quantities against each other in fig. 2a-b.

We have added a new paragraph discussing this point, which reads:
"As these scenarios are concentration-driven rather than emissions-driven, the uncertainty due to carbon cycle processes shows up in figure 3 as a horizontal divergence between ensemble members, rather than a vertical divergence as it would appear in an emissions-driven scenario. However, the self-consistency between the climate and carbon cycles that results from the inferred-emissions approach, as well as the qualitative consistency between the models and the emulator that was used to translate scenario fluxes to atmospheric $CO_2$ concentrations in the scenario specification, together ensure that the behavior will be similar between concentration-driven and emissions-driven dynamics, even under these extreme scenarios with either very high or net negative emissions."

3) Third, about the paper conclusions. They are currently suboptimal and not that clear in terms of what new is found in these simulations. For example, a conclusion that land carbon uptake has many uncertainties – repeated several times - could be written without these simulations at all. This is very clear from ESM runs until 2100, and also can be seen in millennium-scale experiments, see eg Joos et al. (2013). Instead of these qualitatively vague conclusions on uncertainty, could you rather suggest how could we proceed in reducing uncertainty, what could be the main factors: response to droughts, $CO_2$ fertilization, natural vegetation dynamics, land use? I also would suggest to focus on what is common among the models, and not only on differences. This would be helpful for carbon cycle emulators to be used for analysis of other future scenarios.

This is a good point and we will focus the text more on identifying the common responses across the models in revision. We do find that the nature of the uncertainty in the model results changes after 2100, both in terms of globally-integrated absolute ensemble spread as well as in the qualitative aspects of drivers and geographic patterns, and will emphasize this change in the uncertainty in revisions as well. We agree with the suggestion here as well to also focus some additional text on ways that we might focus efforts on reducing the uncertainty.

We have now included more info on how the ensemble spread changes in time and across scenarios, which allow us to describe more quantitatively how the uncertainty shifts. We have also added a figure on the latitude/longitude dynamics to highlight where there is agreement (and what that agreement is) as well as where there is a lack of agreement.

Minor comments

P2., l.44: the long lifetime of $CO_2$ in the atmosphere: I miss here citations of papers focused on this issue, eg Archer et al. (2009), Joos et al. (2013) – please refer to them in the introduction.

We will add these citations in revision.

We have added citations to these two papers here.

Section 2.1: ScenarioMIP protocol doesn't define how land use forcing (including wood and crop harvest) and aerosol forcing are implemented after 2100. Land cover forcing in Hurtt et al. (2020) is provided only until 2100. How is it extended beyond 2100? Are these forcing extensions treated in the same way in all models? These details should be explained.

We will add discussion of these questions in revision.

We have added a sentence here to address this "In the long-term extensions, land use is held constant after 2100, as described in Meinshausen et al., (2020)."

l.114-115. This statement belongs to the Results section, not scenarios (unless scenario forcings were averaged).

We will edit in revision.

We have moved this to the model description, rather than the scenarios subsection, since it details how we analyzed the model output.

Section 2.2 Model descriptions here is extensive and inconsistent among models. It doesn't allow to capture at the first glance a difference between models. Arora et al. (2020) did a better job with their Table 2 summarizing all important details of participating models. Some details like on the permafrost carbon dynamics in CESM could be reported in addition to the new Table.

Also: some descriptions here refer to components which are most likely not used in the study (such as wetland emissions in CanESM2). Why do you need to mention them?

As both reviewers thought that a more systematic description of a smaller set of relevant processes would be helpful here, we will edit in revision to follow this suggestion.

We have moved the model description text to the Appendix A1, and only go over some large-scale features of the models (with a focus on the land models since their behavior is so different and central to the discussion) here.

p. 15, l. 467-472 – very long sentence, cut it in two after the reference. Interesting is the overshoot behavior.

We will edit in revision.

We have split this sentence into two shorter ones.

P.15 – last sentence is unclear, what exactly is meant by reference to paleoclimate? Does it mean that evaluation by paleo runs allows to avoid surprises or other way around?

We will update this in revision.

We have reworded this to say, "That the model satisfies the transient constraint underscores the possibility for nonlinearities in temperature versus cumulative emissions, although the long-term sensitivity may separately be constrained by paleoclimate evidence (Sanderson, 2020; Tierney et al., 2020)."

P16., l. 508 what is meant by budgeting here?

We will clarify this in revision.

We have replaced the word budgeting with accounting.

l. 512 How would you propose to test carbon model dynamics on long timescale, against what evidence? Emitting ca. 5000 PgC in RCP8.5 goes beyond a scale of any available evidence for the last 30 million years, and the quality of data for deep paleo is really poor.

Good points, we will discuss further in revision.

We have decided not to go into further detail on this point. As the reviewer points out, the use of paleoclimate constraints on ESM behaviors is a complex topic, and paleoclimate evidence is subject to its own limitations.  Because we aren't actually doing this work here, we don't want to identify the limits here; the existing text is arguing that the community has not been making full use of paleoclimate constraints that could potentially be done now.

Fig. 2 a-b: dashed lines are hard to see. Why don't you use the same color code for models as on figure 3? It will make figure consistent. Labels on c-d are hard to see on a paper, this figure is not readable on a paper.

Currently the color/dash patterns are consistent between figure 1 and figure 2a-b.  The color/dash patterns used in figure 3 would require separating out the two scenarios into different panels, which we specifically don't want to do here so as to enable comparison of the two scenarios.  We will likely keep the colors and dash patterns in figure 2a-b as they are, but will increase the text size in figure 2c-d so as to improve readability.

We have kept the colors and patterns as-is, but have increased the font sizes throughout to ensure better readability.

Fig. 2: I suggest to add a table with values of emissions, land and ocean net fluxes at the years 2100, 2200, and 2300 for every model and for ensemble-mean. This would help to see a difference between models and time slices.

We will consider this in revision and see if including such a table improves the flow of the discussion.

We have added a table with cumulative carbon fluxes of fossil fuel emissions, land sink, and ocean sink, for each scenario and each time period, both by model and for the ensemble mean.

Fig. 2 caption: using the term "biospheric fluxes" for net land and ocean fluxes is confusing. Ocean carbon uptake is mostly inorganic, unless the authors could disentangle changes in anthropogenic uptakes due to solubility and biology. I suggest consistently use land and ocean uptakes. Strictly speaking, landuse emissions are anthropogenic too, so one rather has to use "fossil-fuel" emission term.

Good points, we will reword these as the review suggests in revision.

We have changed the wording in both the caption and the figure text to specify land and ocean fluxes, and replaced "anthropogenic" with "fossil fuel". We have also changed the temperature vs cumulative emissions plot to include both fossil fuel and land use in the cumulative emissions term, and noted that in the figure caption.

Fig. 3. There is a principal inconsistency between C-driven and E-driven TCRE plots, as explained in the second major comment. This has implications for this plot, as C dynamics of MAGICC are different from the ones of ESMs. A negative betta feedback in E-driven run will likely stabilize the system much faster than in the C-driven simulation. Please discuss it.

While there is certainly an important difference in the interpretation of a TCRE plot between C-driven and E-driven runs, which we can and will discuss in greater detail in revisions, we disagree that it substantially changes the interpretation, as the reviewer suggests here. Basically, model structural and parametric differences related to the carbon cycle generate uncertainty (in the form of inter-model plume spread) that shows up in the horizontal dimension in a C-driven run, whereas the same uncertainty shows up in the vertical dimension in an E-driven run. However, the requirement of self-consistency implies that the overall uncertainty is present in both cases.

We have added a new paragraph making this point, the text of which we pasted above in response to the related earlier comment.

Fig. 4-5. These Hovmöller diagrams are very useful, but they hide the fact that the zonal land distribution is inequal. This is especially valid for the latitudes to the south of 40°S. I suggest to cutoff these diagrams at 40°S or 50°S.

This is a good suggestion, we will consider cropping the terrestrial figures at 40S or 50S to see if it enhances readability of the figures.

We have updated these figures now, so that all land quantities are only between 70N and 50S.

Also - ESMs have 2-D geographic distribution of carbon. You can add a figure of ensemble mean for land and ocean net fluxes at 2300 for two scenarios. It will be very useful to see such a plot corresponding to Fig. 2 c-d.

This is a good suggestion, we will consider adding maps of this, possibly broken out to distinguish integrated changes before and after 2100, and with hatching or similar to indicate the degree of model agreement on the sign of response.

We have added a new figure, now figure 7, which shows maps of the carbon change for the two scenarios over several intervals: historical, 2015-2100, 2100-2200, and 2200-2300, for both of the scenarios analyzed here, and show the ensemble-mean value and the hatching for model agreement. We have al;so added a new section, 3.5.3, which discusses the results shown in the figure.

**References**

Archer, D., Eby, M., Brovkin, V., Ridgwell, A., Cao, L., Mikolajewicz, U., Caldeira, K., Matsumoto, K., Munhoven, G., Montenegro, A., Tokos, K., 2009. Atmospheric lifetime of fossil-fuel carbon dioxide, Annual Reviews of Earth and Planetary Sciences, 37, 117-134.

Joos, F., Roth, R., Fuglestvedt, J. S., Peters, G. P., Enting, I. G., von Bloh, W., Brovkin, V., Burke, E. J., Eby, M., Edwards, N. R., Friedrich, T., Frölicher, T. L., Halloran, P. R., Holden, P. B., Jones, C., Kleinen, T., Mackenzie, F. T., Matsumoto, K., Meinshausen, M., Plattner, G.-K., Reisinger, A., Segschneider, J., Shaffer, G., Steinacher, M., Strassmann, K., Tanaka, K., Timmermann, A., and Weaver, A. J.: Carbon dioxide and climate impulse response functions for the computation of greenhouse gas metrics: a multi-model analysis, Atmos. Chem. Phys., 13, 2793–2825, https://doi.org/10.5194/acp-13-2793-2013, 2013.

---

## Author Response (AR2)

We thank both referees as well as the editor for their time and care in handling this manuscript. In this point-by-point response, our original responses to the referees have been left in blue text, and the new responses with specific changes are in green.

We again thank the referees and editor for their careful consideration of the manuscript. Our updated January 2022 responses are in magenta. We further address the reviewer 2 comment 2, and in the manuscript, we have also added a new figure 4 and discussion about this figure, demonstrating that the overshoot asymmetry can be predicted by a model's previously-published reported values of ZEC. This additional figure relates to that specific reviewer's comment, as it shows that there is a high degree of consistency between emissions-forced and concentration-forced ESM runs, even under negative emissions, as well as a consistency in Earth system dynamics between zero and negative $CO_2$ emissions. We thank both the reviewer and editor for the opportunity to respond further to this point, which we believe substantially strengthens the argument in the manuscript.

Reviewer 2 (Victor Brovkin) Comments:

2) Second, there are limitations of concentration-driven runs in analysis of TCRE (Fig. 3). Emissions in these runs are not purely anthropogenic but ESM-inferred emissions. Using monotonously increasing $CO_2$ scenarios is all right, but when concentrations start to decline, one can do wrong conclusions about temperature-cumulative emissions relationship if interactive carbon cycle is ignored. It is counter-intuitive: after cessation of fossil-fuel emissions in IAM scenarios, ocean naturally continues to take carbon, and ESM-inferring approach would count this ocean uptake as continuing anthropogenic emissions. Really confusing! This limitation must be discussed in depth for Fig. 3.

We will add further discussion of this in revision. However, the point that the reviewer makes here isn't necessarily true, so long as the climate emulator used by the IAM that prescribes the concentration trajectory for a given scenario is doing a good job of emulating the Earth system. For example, in the zero-emissions condition that the reviewer refers to, the atmospheric $CO_2$ concentration will decline in accordance with the continued ocean carbon uptake. Because ESM-inferred emissions are calculated based on the change in time in the integrated carbon summed across the land, ocean, and atmosphere together, an atmosphere-to-ocean carbon flux should not lead to any inferred anthropogenic emissions at all. If the IAM and ESM were to disagree on the relative amount of ocean and land uptake in response to the forcings, then the atmospheric $CO_2$ concentration would not reflect the ESM fluxes, and the inferred emissions from the ESMs would deviate from those generated by the IAM and used to construct the scenario, and this is why we compare these quantities against each other in fig. 2a-b.

We have added a new paragraph discussing this point, which reads:

"As these scenarios are concentration-driven rather than emissions-driven, the uncertainty due to carbon cycle processes shows up in figure 3 as a horizontal divergence between ensemble members, rather than a vertical divergence as it would appear in an emissions-driven scenario. However, the self-consistency between the climate and carbon cycles that results from the inferred-emissions approach, as well as the qualitative consistency between the models and the emulator that was used to translate scenario fluxes to atmospheric $CO_2$ concentrations in the scenario specification, together ensure that the behavior will be similar between concentration-driven and emissions-driven dynamics, even under these extreme scenarios with either very high or net negative emissions."

In the newest version, we have added further detail on this point. In the methods section, we have added another paragraph detailing the logic:

Since the method for inferring compatible fossil fuel emissions from a concentration-driven ESM simulation is based only on conservation of mass, it is equally valid for net positive and net negative CO2 emissions scenarios. However, if the ESMs disagree on the rate of land or ocean carbon uptake with the representation of land and ocean carbon uptake in MAGICC7 used to construct the CO2 concentration timeseries, this disagreement will result in differences between the ESM-inferred and the scenario-specified CO2 emissions. By comparing the ESM-inferred and scenario-specified emissions, we can determine whether any systematic differences between the ESM and MAGICC7 net carbon sinks exist.

Also, we have added another sentence to the paragraph that we had added above, noting that the fact that the ZEC, which is based from emissions-forced runs, predicts the asymmetry in the temperature-cumulative emissions curve, where the emissions are diagnosed from concentration-forced runs, for the overshoot scenario provides further evidence that this comparison can be made.

The consistency between the model dynamics that are concentration-forced here and those of the emissions-forced runs from ZECMIP (MacDougall et al., 2020) further supports the argument that temperature-cumulative emissions relationships between concentration-forced and emissions-forced experiments are comparable even under strong net negative $CO_2$ emissions.